

# High-time Resolution Source Apportionment of PM$_{2.5}$ in Beijing with Multiple Models

Yue Liu [1], Mei Zheng [1, *], Mingyuan Yu [1], Xuhui Cai [1], Huiyun Du [2,3], Jie Li [2], Tian Zhou [1], Caiqing Yan [1], Xuesong Wang [1], Zongbo Shi [4,5], Roy M. Harrison [4,6], Qiang Zhang [7], Kebin He [7]

[1] SKL-ESPC and BIC-ESAT, College of Environmental Sciences and Engineering, Peking University, Beijing 100871, China

[2] State Key Laboratory of Atmospheric Boundary Layer Physics and Atmospheric Chemistry, Institute of Atmospheric Physics, Chinese Academy of Sciences, Beijing 100029, China

[3] Center for Excellence in Urban Atmospheric Environment, Institute of Urban Environment, Chinese Academy of Sciences, Xiamen, China

[4] Division of Environmental Health and Risk Management, School of Geography, Earth and Environmental Sciences, University of Birmingham, Edgbaston, Birmingham, B15 2TT, UK

[5] Institute of Surface Earth System Science, Tianjin University, Tianjin, 300072, China

[6] Department of Environmental Sciences / Center of Excellence in Environmental Studies, King Abdulaziz University, PO Box 80203, Jeddah, 21589, Saudi Arabia

[7] State Key Joint Laboratory of Environment Simulation and Pollution Control, School of Environment, Tsinghua University, Beijing 100084, China

*Correspondence to:* Mei Zheng (mzheng@pku.edu.cn)

## Abstract

Beijing has suffered from heavy local emissions as well as regional transport of air pollutants, resulting in severe atmospheric fine particle (PM$_{2.5}$) pollution. This study developed a combined method

to investigate source types of PM2.5 and its source regions during winter 2016 in Beijing, which include receptor model, footprint, and an air quality model. The receptor model was performed with high-time resolution measurements of trace elements, water soluble ions, organic carbon, and elemental carbon using online instruments during the wintertime campaign of the Air Pollution and Human Health-Beijing





(APHH-Beijing) program in 2016. Source types and their contributions estimated by the receptor model

(Positive Matrix Factorization, PMF) using online measurement were linked with source regions identified by the footprint model, and the regional transport contribution was estimated by an air quality model (the Nested Air Quality Prediction Model System, NAQPMS) to analyze the specific sources and source regions during haze episodes. Our results show that secondary and biomass burning sources were dominated by regional transport while the coal combustion source showed an increased local contribution,

suggesting that strict control strategies for local coal combustion in Beijing and a reduction of biomass burning and gaseous precursor emissions in surrounding areas were essential to improve air quality in Beijing. The combination of PMF with footprint results revealed that the secondary source was mainly associated with southern footprints. The northern footprint was characterized by a high dust source contribution (11%) while industrial sources increased with the eastern footprint. The results demonstrated

the power of combining receptor model-based source apportionment with other models in understanding the formation of haze episodes and to identify specific sources from different source regions affecting air quality in Beijing.

Keywords: Source apportionment, multiple models, regional transport


## 1. Introduction

Presently, haze in China has the characteristics of high frequency and long duration on a regional scale, and has influenced public life and human health. High concentrations of fine particulates, which can significantly reduce atmospheric visibility, are one of the main factors in the formation of haze

episodes (Sun et al., 2016b; Watson et al., 2002; Yang et al., 2015). Previous studies have found that $PM_{2.5}$ can be emitted from various sources, including coal combustion, biomass burning, traffic sources, industrial sources and dust (Watson et al., 2001; Gao et al., 2016). Therefore, it is important to have a better understanding of the major source types and their contribution to $PM_{2.5}$ in order to formulate



science-based effective policies and regulations.

As the capital of China, Beijing has suffered from heavy emissions from various sources, resulting in severe PM$_{2.5}$ pollution (Li et al., 2017a; Lv et al., 2016). The source apportionment of PM$_{2.5}$ in Beijing has received great attention in recent years, which is mostly based on receptor models (Gao et al., 2016; Li et al., 2017d; Lv et al., 2016; Song et al., 2006; Yang et al., 2016). Receptor models including the Chemical Mass Balance model (CMB) and Positive Matrix Factorization model (PMF) are the most

commonly used methods of source apportionment in China, and are implemented by application of mathematical methods to measurements of chemical composition of fine particles at receptor sites (Cooper et al., 1980; Gao et al., 2016; Lv et al., 2016; Zheng et al., 2005). The model can identify and quantify the contribution of multiple source types based on in-situ measurements and specific source tracers. Gao et al. (2016) employed two receptor models, PMF and Multilinear Engine 2 (ME2), to

conduct high-time resolution source apportionment of PM$_{2.5}$ in summer in Beijing. The results showed that PMF and ME2 corresponded well with each other, and secondary sources were predominant in Beijing (38-39%). Similar source apportionment results were reported by in Peng et al. (2016) with secondary sources contributing 35-40%. Sun et al. (2016b) used online instruments and PMF to investigate the rapid evolution of a severe haze episode in winter in Beijing and showed the variation of

chemical components during four stages of haze. By conducting receptor models based on high-time resolution online measurement, the source types and source contributions in Beijing have been analyzed in previous studies. However, these studies still have limitations in that source apportionment based on receptor models are only restricted to one or several receptor sites without information about detailed source regions as well as the local and regional source contributions.

Previous studies have indicated that PM$_{2.5}$ pollution in Beijing has been significantly influenced by regional transport and meteorological conditions (Han et al., 2015; Li et al., 2017a; Zhao et al., 2013). With the development of the function of source apportionment in air quality models, source regions and relative contributions to the receptor site could be quantitively estimated, based on emission inventory of





pollution sources and meteorological fields (Burr et al., 2011; Kwok et al., 2013; Zhang et al., 2015). Li
et al. (2016) found that regional transport highly contributed to the rapid increase stage, with the transport
height ranged from 200 to 700 m above ground level with application of the Nested Air Quality Prediction
Model System (NAQPMS). Han et al. (2018) used a regional air quality modeling system coupling with
ISAM (Integrated Source Apportionment Method) and found that air pollutants derived from Hebei and
Shandong provinces were major contributors to PM$_{2.5}$ in Beijing, with contributions up to 25% and 10%,
respectively. The air quality model has advantages of analyzing spatial distribution and regional transport
of pollutants, but it also has large uncertainties due to emission inventory, boundary layer meteorological
processes and complex atmospheric chemical processes.

Due to the importance of the regional transport contribution to PM$_{2.5}$ in Beijing, the limitations of
receptor models cannot be ignored. The source types and source contribution derived from receptor
models can be combined with the contribution and direction of regional transport derived from chemical
transport models. In this study, we employed the receptor model (PMF), the air quality model (NAQPMS)
and a footprint model simultaneously based on high-time resolution online measurement data to
investigate sources and regional transport of PM$_{2.5}$ in Beijing during November to December in 2016, as
part of the Air Pollution and Human Health (APHH) campaign. The goal of the study is to link the
contribution of different sources by PMF with the source regions by footprint, and the regional transport
contribution by NAQPMS. The combination of multiple models gives greater power to identify specific
sources and source regions.

## 2. Materials and methods

### 2.1 Online measurement of PM$_{2.5}$

Online sampling of PM$_{2.5}$ was conducted from November 2016 to December 2016 in winter, which
was within the heating period of Beijing. The sampler was operated at the Peking University monitoring
site (PKU, 39°59′21″N, 116°18′25″E) in the northwestern part of Beijing city. There are no obvious





emission sources locally, except two major roads (150 m to the east and 200 m to the south). Situated in

a mixed district of teaching, residential, and commercial areas, the sampling site is representative of the

Beijing urban area. The room is located on the sixth floor of a teaching building within PKU. The inlet of

the instrument is about 20 m above the ground.

Online $PM_{2.5}$ mass concentrations were continuously measured using a Tapered Element Oscillating

Microbalance (TEOM 1405F, Thermo Fisher Scientific Inc.). Organic carbon (OC) and elemental carbon

(EC) were simultaneously monitored by a Semi-continuous OCEC Carbon Aerosol Analyzer (Sunset

Laboratory Inc.) with 1-h time resolution.

An in-situ Gas and Aerosol Compositions monitor (IGAC, Model S-611, Fortelice International

Co.Ltd.), which could collect both gases and particles simultaneously, was applied to measure water-

soluble ions online with 1-h time resolution in this study. A detailed description of IGAC can be found in

Young et al. (2016). Briefly, IGAC was composed of three major units, including a Wet Annular Denuder

(WAD) to collect gases into aqueous solution, a Scrub and Impact Aerosol Collector (SCI) to collect

particles into solution and a sample analysis unit comprised of two ion chromatographs (DionexICS-1000)

for analyzing anions and cations (IC). Ambient air was drawn through a $PM_{10}$ inlet followed by a $PM_{2.5}$

cyclone at a flow rate of 16.7 L min$^{-1}$, and then gases and $PM_{2.5}$ were separately collected by WAD and

SCI. Both gaseous and aerosol samples were injected into 10 mL glass syringes which were connected to

the IC for analysis (30-min time resolution for each sample). The concentrations of eight water-soluble

inorganic ions (e.g., $NH_4^+$, $Na^+$, $K^+$, $Ca^{2+}$, $Mg^{2+}$, $SO_4^{2-}$, $NO_3^-$ and $Cl^-$) in the fine particles were measured.

Twenty-three trace elements in $PM_{2.5}$ were measured by an Xact 625 Ambient Metal Monitor

(Cooper Environmental Services LLC, USA) with 1-h time resolution. Among them twelve elements (e.g.,

K, Ca, Ba, Cr, Mn, Fe, Cu, Ni, Zn, As, Se, Pb) were selected for further analysis, while other trace

elements (such as V, Co, Tl) were not used here due to the low concentrations (below the method detection

limit). The ambient air was sampled on a Teflon filter tape inside the instrument through a $PM_{2.5}$ cyclone

inlet at a constant flow rate of 16.7 L min$^{-1}$, and then the sample was automatically analyzed by



nondestructive energy-dispersive X-ray fluorescence (XRF) to determine the mass of the species. This

instrument has been documented with Environment Technology Verification (ETV) and certified by the

US Environment Protection Agency (EPA, 2012).

Strict quality assurance (QA) and quality control (QC) protocols for online instruments were

performed during the whole sampling period. For IGAC, the internal standard (LiBr) was added

continuously to each sample and analyzed by the IC system during the analysis to check the stability of

the IGAC instrument. During the sampling period, the mean concentrations of $Li^+$ and $Br^-$ were within

the range of three standard deviations, suggesting a stable condition of the IGAC. As shown in Fig. S1,

the slope of the linear fitting between the anions and cations was 0.93, and $R^2$ was 0.96. As for the OC/EC

analyzer, external standard calibration using the stock sucrose solution was conducted before operation

to calibrate carbon analysis. For the Xact, a Pd rod was used as automatic internal quality control to check

the performance of the instrument on a daily basis. Additionally, a QA energy calibration test and QA

energy level test were performed for a half hour after midnight every day to monitor any possible shift

and instability of the XRF. During our field campaign, the Xact remained stable and reliable.

### 2.2 Methodology

### 2.2.1   Positive Matrix Factorization (PMF)

To qualitatively and quantitatively identify sources of $PM_{2.5}$ and estimate the associated

contributions, the USEPA PMF 5.0 model was adopted in this study. The principle and detailed

information on this model may be found in Paterson et al. (1999) and the EPA 5.0 Fundamentals and User

Guide. Factor contributions and profiles were derived by minimizing the objective function Q in the PMF

model, which was determined as follows:

$$Q = \sum_{i=1}^{n} \sum_{j=1}^{m} \left[ \frac{x_{ij} - \sum_{k=1}^{p} g_{ik} f_{kj}}{u_{ij}} \right]^2 \tag{1}$$

Data values below the MDL were substituted with MDL/2. Missing data values were substituted



with median concentrations. If the concentration was less than or equal to the MDL, the uncertainty (Unc) was calculated using a fixed fraction of the MDL:

$$\text{Unc} = \frac{5}{6}MDL \tag{2}$$

If the concentration was greater than the MDL provided, the calculation was based on the following equation:

$$\text{Unc} = \sqrt{(Error\ Fraction \times concentration)^2 + (0.5 \times MDL)^2} \tag{3}$$

In total, nineteen chemical components were used in the PMF model, including OC, EC, Cl$^-$, SO$_4^{2-}$, NO$_3^-$, Na$^+$, NH$_4^+$, K, Ca, Ba, Cr, Mn, Fe, Cu, Ni, Zn, As, Se and Pb. To determine the optimal number of source factors, a string of effective tests, in which factor number was from four to eight, was carried out and among which the best performance was found at six factors with the lowest $Q_{Robust}$ and $Q/Q_{expected}$ value. Bootstrapping (BS), displacement (DISP), and bootstrapping with displacement (BS-DISP) were conducted to analyze the uncertainty of the PMF model at six factors. The results were stable with all factors mapped in BS in 100% and no swaps with DISP and all BS-DISP runs, indicating a convincing source apportionment result.

### 2.2.2   Footprint analysis model

The footprint model developed by Peking University was used to simulate the potential source region of air pollution. The footprint is a transfer function in a diffusion problem linking the source and the measurement result at a point (receptor) (Pasquill and Smith 1983). That is,

$$c(r) = \int_R Q(r + r')f(r, r')dr' \tag{4}$$

where c is the measured concentration at spatial location  r,  Q  is the source strength with spatial location $(r + r')$, $f$  is the footprint or the transfer function and $R$ is the integration domain. The footprint links point measurements (receptors) in the atmosphere to upstream forcings, in which turbulent dispersion plays a central role. The Lagrangian stochastic (LS) particle models was used to calculate the footprint



function (Cai et al., 2007; Leclerc and Thurtell 1990; Kurbanmuradov and Sabelfeld 2000).

The meteorological data used to drive footprint model was provided by the Weather Research and Forecasting model (WRF-ARWv3.6.1) (http://www.wrf-model.org/), initialized using the Final Analysis (FNL) data from the United States National Centers for Environmental Prediction (NCEP). Two nested domains were used in this study with horizontal resolutions of 15 and 5 km and 28 vertical levels. The simulation period was from November 1 to December 31, 2016, with a 12 h spin-up time before the start for each 48 h simulation. The domain of the footprint model is the same as the domain 2 in WRF which covers the North China Plain (500×600km), and the horizontal resolution is 2.5 ×2.5 km. Every hour, 5000 particles were released 10 m above the ground at the center of Beijing, and then each particle was tracked backward in time for 48 hours. The residence time of all particles in 0-100 m above the ground were recorded to obtain the footprint. This model has undergone rigorous theoretical discussion and verification and more detailed principle and calculation methods of the model can be found in Cai et al. (2007).

### 2.2.3 The Nested Air Quality Prediction Model System (NAQPMS)

In this study, the NAQPMS model was applied to analyze the contribution of local emissions and regional transport to $PM_{2.5}$ in winter in Beijing. NAQPMS is a 3-D Eulerian chemical transport model with terrain-following coordinates, developed by the Institute of Atmospheric Physics, Chinese Academy of Sciences (IAP/CAS) and has been validated by the Ministry of Environmental Protection of China (CMEP, 2013). The main modules in the model include horizontal and vertical advection and diffusion, dry and wet deposition, and gaseous, aqueous, aerosol and heterogeneous chemistry (Li et al., 2007; Li et al., 2017b). A more detailed description of the model can be found in Li et al. (2008; 2014; 2016; 2017b).

Three nested model domains were used in this study. The coarsest domain (D1) covered most of China and East Asia with a 27 km resolution. The second domain (D2) included most anthropogenic emissions within the North China Plain with a 9 km resolution. The innermost domain (D3) covered the



Beijing-Tianjin-Hebei region at a 3 km resolution. The first level of model above the surface is 30 m in height, and the average vertical layer spacing between 30 m and 1 km is around 100 m. The MIX (http://www.meicmodel.org/dataset-mix.html) anthropogenic emission inventory was used (Li et al.,

2017c), with the original resolution of 0.25 °(about 25 km at middle latitudes) and the year of 2010. The NAQPMS meteorological fields was provided by the Weather Research and Forecasting model (WRF-ARWv3.6.1) (http://www.wrf-model.org/) driven by the National Centers for Environmental Prediction (NCEP) Final Analysis (FNL) data. The simulation was conducted from November 10 to December 15, 2016.


### 2.2.4 The combination of multiple models

The footprint model was used to provide the direction of source regions while the NAQPMS model was run to calculate the contribution of local emission and regional transport. To verify the consistency between the two models, the footprint with a time resolution of 6 hours was divided into four types (local,

south, north and east) according to the direction of potential source regions, and average local contributions of different types obtained from NAQPMS were calculated (See Table S1). A typical example of different types of footprint can be seen in Fig. S2. The average local contribution estimated by NAQPMS was highest for the local footprint (85%) and lower for south (68%), north (63%) and east footprints (66%). The results of the two models correlated well with each other.

Based on online measurement of $PM_{2.5}$ species including specific source tracers, the receptor model (PMF) could be used to obtain precise source apportionment result but with no information upon regional transport. Therefore, the footprint and NAQPMS model were simultaneously conducted and combined with the PMF model to link the source type and contribution to $PM_{2.5}$ in Beijing by receptor models with different source regions.


### 3. Results and discussion



### 3.1 Mass concentration and chemical composition of PM$_{2.5}$

Temporal variation of chemical composition of PM$_{2.5}$ during the field campaign was shown in Fig. 1. Organic matter (OM) was calculated as OM= 1.8 ×OC (Pitchford et al., 2007). Mineral species was calculated as Mineral= 1.89 Al +2.14 Si + 1.4 Ca + 1.43 Fe + 1.66 Mg (Zhang et al., 2003). The concentrations of Al, Si, Fe and Mg were calculated by the concentration of Ca and the composition of urban soils of Beijing: Al= 1.7Ca, Si= 7.3Ca, Fe=0.7Ca, Mg= 0.3Ca (An et al., 2016). Since the concentration of Al and Si were not directly measured by Xact, the calculated mineral component might be underestimated. "Others" were calculated by subtracting OM, EC, Mineral and secondary inorganic aerosol (SIA, including SO$_4^{2-}$, NO$_3^-$, NH$_4^+$) concentration from total PM$_{2.5}$ concentration. Figure 1 shows that SIA and OM were the predominant PM$_{2.5}$ components in winter in Beijing, accounting for 55% and 27% of total PM$_{2.5}$ mass, respectively. The average concentration of OC was 20. 8±17.0 μg m$^{-3}$, and the average concentration of EC was 5.59 ± 4.43 μg m$^{-3}$. The OC/EC ratio is often used to indicate the contribution of primary emission sources and secondary organic aerosols (SOA) (Lim et al., 2002; Zheng et al., 2014). In this study, the OC/EC ratio ranged from 1.36 to 7.92 with an average ratio of 3.91±0.91, which was lower than that in the winter of Beijing in 2013 (5.73 ± 2.47) (Yan et al. 2015). SO$_4^{2-}$ is the predominant ion in SIA with an average concentration of 23.5±20.8 μg m$^{-3}$, which was higher than that of NO$_3^-$ (22.0±23.3 μg m$^{-3}$). The concentration of elemental components ranked from high to low as K> Fe> Ca> Zn> Pb> Mn> Ba> Cu> As> Cr> Se> Ni, with K contributing 2% to PM$_{2.5}$. In general, the large contribution of SIA, OM as well as the high OC/EC ratio indicated the importance of secondary formation in winter in Beijing, while the high concentration of species like SO$_4^{2-}$ and K suggested a significant contribution of combustion sources to PM$_{2.5}$.

Figure 2 shows the large differences in chemical composition of PM$_{2.5}$ concentration between non-haze and haze episodes. Under low PM$_{2.5}$ concentration (< 50 μg m$^{-3}$), the contribution of sulfate increased significantly (up to 24%). When PM$_{2.5}$ was from 50~150 μg m$^{-3}$, OM was the dominant composition (about 40%). When PM$_{2.5}$ was greater than 150 μg m$^{-3}$, the contribution of SIA increased with the





concentration level (up to 55%). The contribution of mineral components decreased from 8% to 2% when PM$_{2.5}$ concentration increased from below 50 μg m$^{-3}$ to over 250 μg m$^{-3}$. The proportion of K, Pb, As and Se, which were tracers of biomass burning and coal combustion, increased with PM$_{2.5}$ concentration. While the contribution of Ca, Ba, Fe, tracers of dust source, decreased with PM$_{2.5}$ concentration. Taken together, all these variations of source specific PM$_{2.5}$ compositions suggested more significant influence of combustion sources to PM$_{2.5}$ in haze episodes and relatively higher contribution of dust source in non-haze periods.

## 3.2 Source apportionment during haze and non-haze periods

To conduct high-time resolution source apportionment in Beijing, a PMF model was applied to 1-h online measurement data. The six-factor solution gave the best performance. The factor profile for each factor is shown in Fig. S3. Contribution of different factors to PM$_{2.5}$ were estimated after considering major sources in Beijing, the similarity of the PMF source profiles with relevant source emission profiles, and distinctively different marker species for different sources. Factor 1 was heavily weighted by secondary inorganic ions (SO$_4^{2-}$, NO$_3^-$ and NH$_4^+$) and moderately weighted by OC, which was typical of the secondary source profiles (Gao et al., 2016; Peng et al., 2016; Shi et al., 2017). Factor 2 was highly loaded on metal species including Mn, Fe, Cu and Zn, which were mostly used as indicators for industrial sources (Hu et al., 2015; Li et al., 2017; Pan et al., 2015; Yu et al., 2013). Factor 3 presented high loading of Ca, Ba and Fe, and the two crustal elements were mainly emitted from dust sources (Amato et al., 2013; Shen et al., 2016). Factor 4 was mostly loaded by EC, OC and moderately loaded by Cu and Zn, which were mainly emitted from lubricant additive of vehicles (Kim et al., 2003; Tao et al., 2014) and wear of brake and tyre (Pant and Harrison, 2013; Perez et al., 2010). High loading of As, Se and moderate loading of OC, EC were observed in Factor 5, indicating a typical source profile of coal combustion (Vejahati et al., 2010). Factor 6 was characterized by high loading of K, SO$_4^{2-}$ and OC, which were identified as indicators of biomass burning (Duan et al., 2004). In previous studies, cooking source could be one of the





important sources of PM$_{2.5}$ (Sun et al., 2013), but in this study cooking source was not identified by PMF due to the lack of organic tracers.

The source apportionment result of PMF in winter in Beijing was shown in Fig. 3. During the campaign, the source contribution in Beijing ranked as secondary source (44%) > traffic source (18%) > coal combustion (16%) > biomass burning (9%) > industrial source (8%) > dust (5%). The high contribution of secondary sources in winter was similar to previous studies (Gao et al., 2016; Peng et al., 2016; Zhang et al., 2013), which might be attributed to regional transport and heterogeneous reactions (Ma et al., 2017).

Considering data integrity and representativeness, four typical pollution episodes (EP1-4) and two non-haze periods (NH1 and NH2) were selected. The average PM$_{2.5}$ concentrations in four haze episodes were all above 75 μg m$^{-3}$ (see Table 1). EP1 (Nov. 14-19) and EP2 (Nov. 24-27) represented the pollution episodes in November, and EP3 (Dec. 1-5) and EP4 (Dec. 16-21) were two severe pollution processes in December. The four pollution episodes were characterized by low wind speed around 2 m s$^{-1}$ and high

relative humidity (RH) compared to non-haze periods (see in Table 1). The chemical composition and sources of the four pollution episodes varied from each other, but relatively high contribution of secondary sources was observed in all episodes (32-57%), and the contribution increased with PM$_{2.5}$ concentration (see Fig. 2). EP4 was characterized by the highest contribution of secondary source (57%). The contribution of coal combustion and industrial sources in EP1 was the most significant compared with

the other episodes, which were 22% and 17% respectively. The traffic source contribution in EP2 and EP3 was higher than other pollution episodes, accounting for about 22%. The source contribution in non-haze periods is significantly different from that in pollution episodes. The contribution of secondary sources in the two non-haze periods, NH1 (Nov. 22-23) and NH2 (Dec. 13-15), decreased to 18% and 25%, while traffic and dust source contribution to PM$_{2.5}$ increased to about 30% and 10%, which could

be influenced by local emission and regional transport from northern areas to Beijing.

Generally, secondary source was predominant (~50%) to PM$_{2.5}$ in pollution episodes, while traffic





source (~30%) became more important in non-haze periods. However, source contribution of PM$_{2.5}$ could vary from episode to episode. EP1 was more influenced by primary sources while EP4 was characterized by high secondary source contribution (57%).


## 3.3 Evolution of different types of haze episodes

The high-time resolution source apportionment result by PMF was combined with the NAQPMS and footprint modelling outcomes to investigate the variation of source types and contributions with source regions in different haze episodes in Beijing. EP1 and EP4, with the longest duration and

significantly different source compositions, were selected as two case episodes for further analysis.

### 3.3.1 A haze episode dominated by local emission

Figure 4 shows the variation of sources and local contribution and Fig. 5 shows the footprint regions and daily source apportionment results by PMF in EP1. It can be seen that EP1 was characterized with

high local contribution (69%-89%) and primary source contribution to PM$_{2.5}$. On November 14, the footprint located in the northeastern part to Beijing (mainly Inner Mongolia) with low PM$_{2.5}$ concentration while the contribution of dust source was significant (52%). On November 16 when the formation stage of EP1 started, the footprint concentrated in local areas of Beijing and the local contribution by NAQPMS (82%) increased simultaneously. The daily average source contribution ranked as traffic source (29%) >

coal combustion (28%) > industrial source (15%) > dust and secondary source (12%) > biomass burning (6%). The contribution of primary sources especially for traffic source increased when footprints were primarily located in local area.

The relationship between source apportionment (1 h) and the footprint model (6 h) could also be found in daily variation of November 17 (see Fig. 5). From 01:00 to 12:00 of the day, the footprint

remained in local areas while primary sources were predominant. However, with the footprint changed to southwestern areas to Beijing from 13:00 to 18:00, the contribution of secondary sources increased

significantly to 42%. After the footprint changed back to local type from 19:00 to 24:00, the secondary source contribution decreased to previous level (19%).

### 3.3.2 A haze episode dominated by regional transport

Figure 6 shows the variation of sources and Fig. 7 shows the footprint regions and daily source apportionment results by PMF in EP4. Different from EP1, the footprint in EP4 was mostly located in the southwestern area to Beijing, where there were heavy polluted cities including Baoding and Shijiazhuang (see Fig. 7). The daily local and regional contribution by NAQPMS of this episode was not provided due to lack of data. From the formation stage (December 16 -17) to the peak (December 20) of EP4, the contribution of secondary sources increased from 34% to 58%, while the contribution of coal combustion and biomass burning were also significant among primary sources (see Fig. 6). Figure 7 shows that the footprint on December 17 was more concentrated in local and eastern areas to Beijing, while it gradually moved to southwestern areas along with the increase of PM$_{2.5}$ concentration and the secondary source contribution.

The above results confirmed that high-time resolution source apportionment result could be integrated with footprint and NAQPMS model to identify the rapid evolution of different episodes - EP1 was an episode mainly caused by local emission from transportation and coal combustion while EP4 was typical for regional transport from southwestern areas to Beijing with increasing contributions of secondary sources.

### 3.4 Relationship of PM$_{2.5}$ sources by PMF with regional transport estimated by NAQPMS

In order to further determine the relationship of PM$_{2.5}$ sources by PMF with regional transport in winter in Beijing, the results of PMF and NAQPMS during the whole campaign were combined for analysis.

### 3.4.1 Sources dominated by local emission and regional transport





Receptor models which are used for source apportionment have the limitation that it could not quantify the local or regional transport contribution. Therefore, the receptor model was combined with the chemical transport model NAQPMS to investigate the correlation of source contribution with local/regional transport. As shown in Sect. 3.2, secondary and combustion sources were predominant in haze episodes in Beijing. To better control those major sources in winter, it is essential to determine correlation of source contribution with the contribution of local emission or regional transport. Figure 8 shows the correlations of relative contribution of secondary sources, coal combustion and biomass burning sources by PMF with local contribution by NAQPMS during the sampling period. The results showed that for $PM_{2.5}$ in Beijing, secondary source contribution decreased when local emission was more significant while coal combustion, as a primary combustion source, showed an increasing trend along with local contribution estimated by NAQPMS. Comparing with Fig. 8 (b) and (c), the two primary combustion sources showed opposite relationship with local contribution, indicating that the pollutants from biomass burning were mainly transported from surrounding areas outside of Beijing while those from coal combustion were more influenced by local emission. According to previous studies, biomass burning was an important source in provinces around Beijing including Shandong, Hebei and Inner Mongolia (Khuzestani et al., 2018; Sun et al., 2016a; Zhang et al., 2010; Zhao et al., 2012; Zong et al., 2016). The surrounding provinces and cities of Beijing are shown in Fig. S4. The results suggested that local-emitted coal combustion contributed significantly to $PM_{2.5}$ in Beijing in winter 2016 and the strict control strategies for coal combustion were essential to improve air quality in Beijing. In the meantime, more control of biomass burning and secondary sources in surrounding areas are also needed to mitigate air pollution in Beijing.

### 3.4.2 Sources dominated in different potential source regions

The combination of PMF result with footprint model was used to further identify specific source type and contribution in different source regions. As mentioned in Sect. 2.2.4, the footprint with the



time resolution of 6 hours was divided into four types (local, south, north and east) according to the

direction of potential source regions. The typical examples of different types of footprint were shown in

Fig. S2. The local footprint referred to the cases with source region located within Beijing. The south

footprint mainly covered southwest areas in Hebei province including Baoding, Shijiazhuang and

Xingtai. The north footprint included Zhangjiakou and Inner Mongolia. The east footprint covered the

north part of Hebei such as Tangshan and Qinhuangdao and the south part of Liaoning province. The

local footprint was predominant in winter in Beijing (N=79) with the contribution of 38%, followed by

north and south footprint (N=51, 45 respectively). The amount of east footprint was the lowest in

winter. The average value and box chart of source contribution in four types of footprint during the

whole sampling period were shown in Fig. 9. It could be seen that local footprint was characterized by

traffic (23%) and coal combustion sources (25%), while the contribution of secondary source (26%)

was the lowest among the four types. On the contrary, secondary source was predominant in south

footprint cases with the contribution of 53%, while the contribution of traffic source decreased to 15%.

The results corresponded well with the analysis of two typical episodes in Sect. 3.3. North footprint was

characterized by the highest contribution of dust source (11%), which could be influenced by dust

transported from Inner Mongolia (Hoffmann et al., 2008; Park et al., 2014). East footprint, which

mainly covered heavy industrial areas such as Tangshan and Shenyang, showed higher contribution of

industrial sources (10%) and coal combustion sources (18%). Figure 9 (b) shows that the variation of

source contribution was the smallest in local footprint, indicating a relatively stable local emission of

Beijing, while the source contribution varied more significantly with east footprint.

        The results of PMF and footprint model showed that source contribution in winter in Beijing was

influenced by the potential source regions, and the predominant source could change specifically for

different footprint type, which might suggest that source apportionment and footprint analysis need to

be combined to better control specific sources from different source regions.



## 4. Summary

High-time resolution online measurements of PM$_{2.5}$ were conducted during the APHH winter campaign in Beijing. Considering the limitation of receptor models which could not identify and quantify regional transport, the receptor model PMF was combined with multiple models including NAQPMS and footprint model to analyze the specific sources from different source regions during haze episodes in Beijing. The source apportionment results by PMF during our sampling period showed that secondary source was predominant (~50%) to PM$_{2.5}$ in pollution episodes, while traffic source (~30%) became more important in non-haze periods. Source contribution of PM$_{2.5}$ could vary from episode to episode.

The multiple models were combined to analyze the evolution of two typical pollution episodes in Beijing. The high-time resolution results indicated that source contribution could vary rapidly and significantly with source regions within different type of haze episodes. EP1, with local concentrated footprint and high local emission, was characterized by coal combustion and traffic sources while EP4 with more southwestern footprint was typical for high secondary source contribution. The relationship of PM$_{2.5}$ sources by PMF with regional transport during the whole sampling period was further investigated. As the predominant sources of PM$_{2.5}$ in Beijing, secondary and biomass burning source were more influenced by regional transport while coal combustion source increased with local contribution. The source regions of PM$_{2.5}$ in Beijing were classified into four types and source contribution varied significantly with potential source regions, with traffic source dominated in local footprint, secondary source dominated in south footprint, dust and industrial source increased in north and east footprint, respectively. The results suggested that source contribution of PM$_{2.5}$ in winter, Beijing could change significantly along with the contribution and direction of regional transport. Therefore, the combined use of receptor model, meteorological model and chemical transport model was important to identify specific sources from different source regions.

## Acknowledgements



This study was supported by funding from the National Natural Science Foundation of China (41571130033, 41430646, 41571130035, 91744203 and 41571130034). Z.S. and R.M.H. acknowledge support from UK Natural Environment Research Council (NE/N006992/1 and NE/R005281/1). The authors gratefully thank for the assistance of Jinting Yu in Peking University for maintaining the online instruments in this work.

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



**Tables and Figure Legends**

**Table 1.** PM$_{2.5}$ and meteorological conditions during pollution episodes and non-haze periods.

**Figure 1.** Chemical composition of PM$_{2.5}$ during sampling period (red for sulfate, blue for nitrate, yellow
for ammonium, green for OM, black for EC, pink for mineral, and grey for others).

**Figure 2.** Variation of chemical composition with PM$_{2.5}$ concentration.

**Figure 3.** Source contribution of PM$_{2.5}$ (a) in the whole sampling period and (b) in different pollution
episodes and non-haze periods (yellow for dust source, green for biomass burning, pink for industrial
source, red for coal combustion, black for traffic source, and blue for secondary source).

**Figure 4.** Variation of sources and local contribution during EP1. The above pie charts show the daily
local (Beijing as BJ) and regional contribution (labeled as Others). The pie charts below show the daily
source type and contribution.

**Figure 5.** Source regions by the footprint model and daily source apportionment results by PMF in EP1.

**Figure 6.** Source contribution in EP4. The pie charts show the daily source type and contribution.

**Figure 7.** Source regions by the footprint model (every 6 h) and daily source apportionment results by
PMF in EP4.

**Figure 8.** Correlations of local contribution by NAQPMS with the relative contribution by PMF of (a)
secondary source, (b) coal combustion source and (c) biomass burning source.

**Figure 9.** (a) The average source contribution (in percentage) for each type of footprint, and (b) box chart
of source contribution in four types of footprint during the whole sampling period. N in (a) represents for
the number of cases. The capital letters in (b) stands for the type of footprint (L for local; S for south; N
for north; E for east) and the lowercases stands for different sources (s for secondary source, c for coal
combustion, t for traffic source, i for industrial source, d for dust, and b for biomass burning).




**Table 1**. PM$_{2.5}$ and meteorological conditions during pollution episodes and non-haze periods.

|  | EP1 | EP2 | EP3 | EP4 | Non-haze | Average |
|---|---|---|---|---|---|---|
| PM$_{2.5}$ ($\mu$g m$^{-3}$) | 98 | 144 | 115 | 242 | 11.1 | 110 |
| Wind speed (m s$^{-1}$) | 2.24 | 2.26 | 2.36 | 2.04 | 4.17 | 2.48 |
| Temperature (°C) | 7.48 | 2.94 | 5.36 | 2.63 | -2.05 | 3.42 |
| Relative humidity (%) | 54.5 | 38.2 | 38.8 | 49.4 | 24.1 | 43.5 |
| Pressure (hPa) | 1012.5 | 1016.5 | 1016.3 | 1016.1 | 1027.9 | 1017.4 |







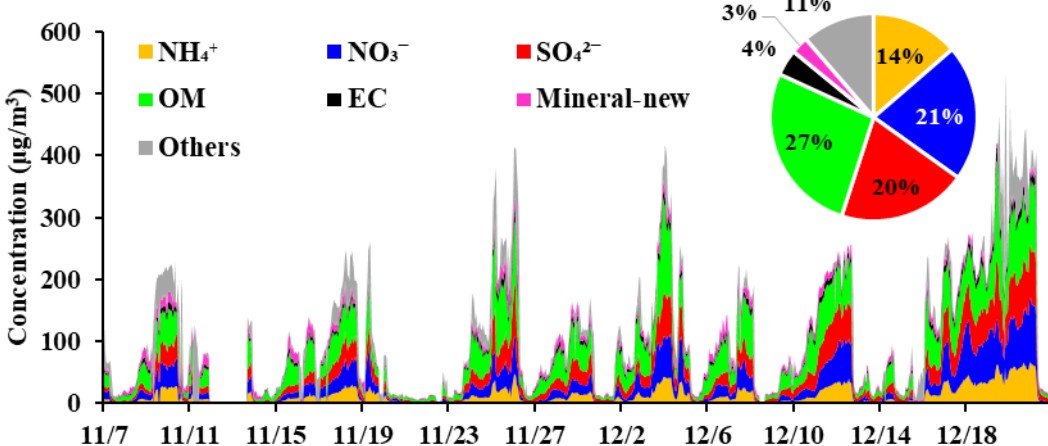

**Figure 1.** Chemical composition of PM$_{2.5}$ during sampling period (red for sulfate, blue for nitrate, yellow for ammonium, green for OM, black for EC, pink for mineral, and grey for others).

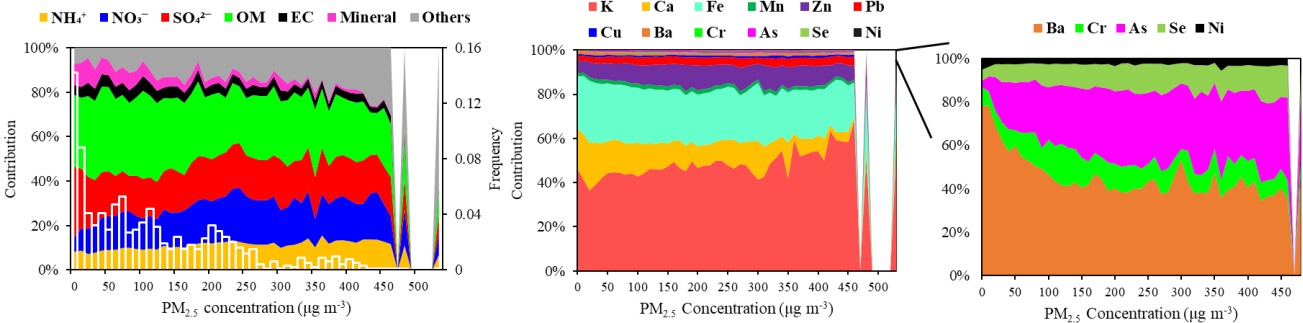


**Figure 2.** Variation of chemical composition with PM$_{2.5}$ concentration.





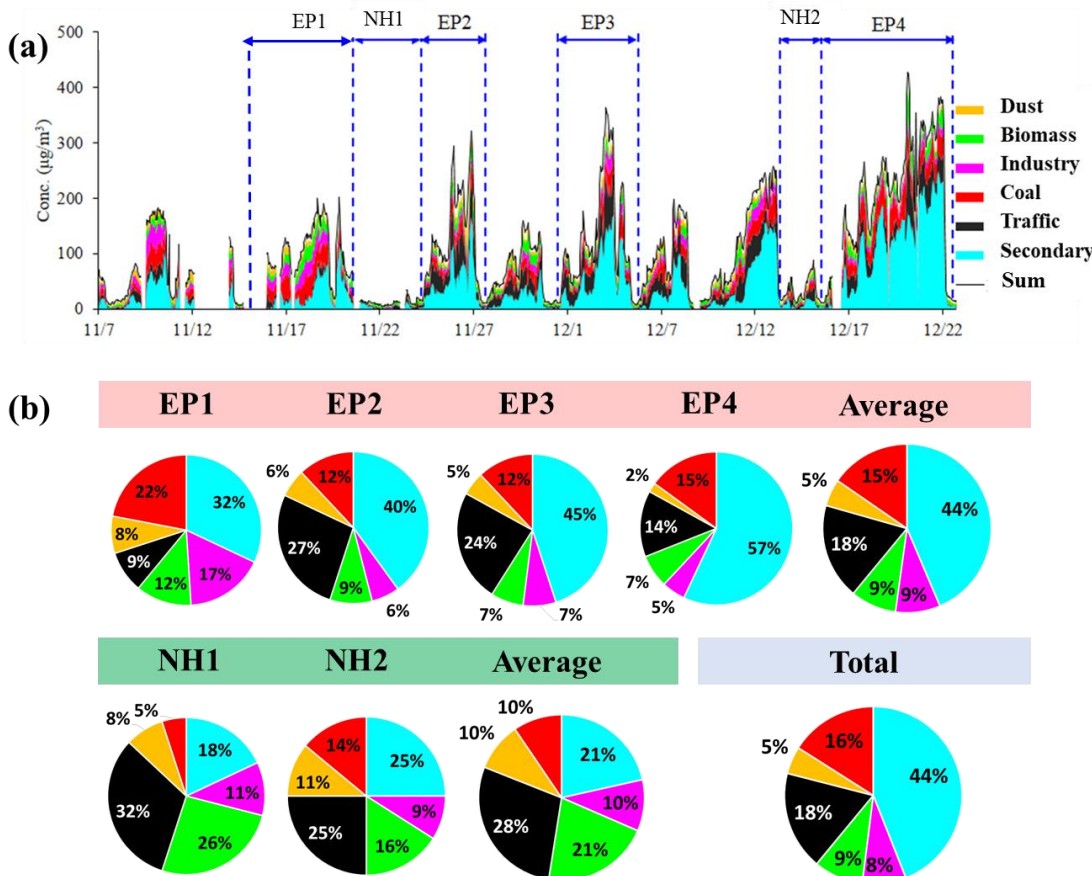

**Figure 3.** Source contribution of PM$_{2.5}$ (a) in the whole sampling period and (b) in different pollution
episodes and non-haze periods (yellow for dust source, green for biomass burning, pink for industrial
source, red for coal combustion, black for traffic source, and blue for secondary source).





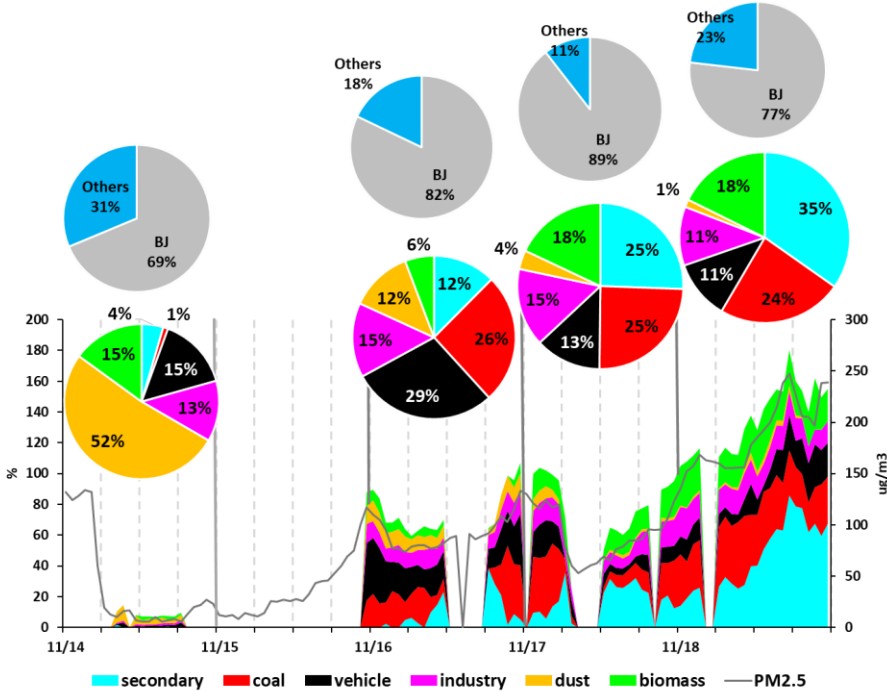

**Figure 4.** Variation of sources and local contribution during EP1. The above pie charts show the daily

local (Beijing as BJ) and regional contribution (labeled as Others). The pie charts below show the daily

source type and contribution.





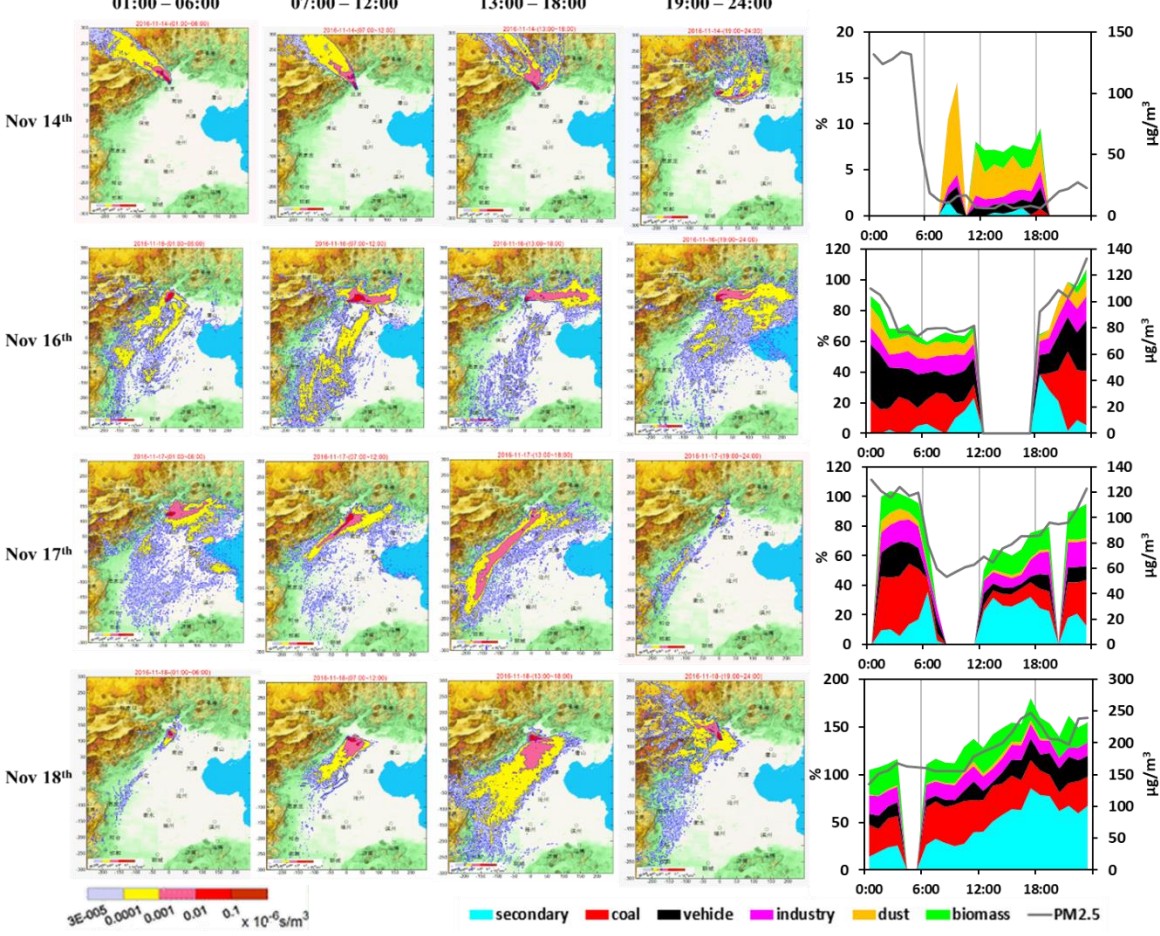

**Figure 5.** Source regions by the footprint model and daily source apportionment results by PMF in EP1.




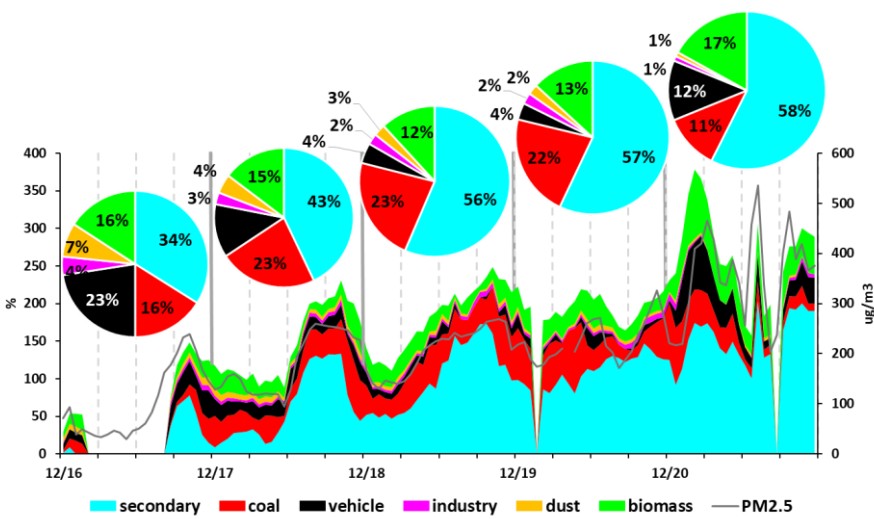

**Figure 6.** Source contribution in EP4. The pie charts show the daily source type and contribution.





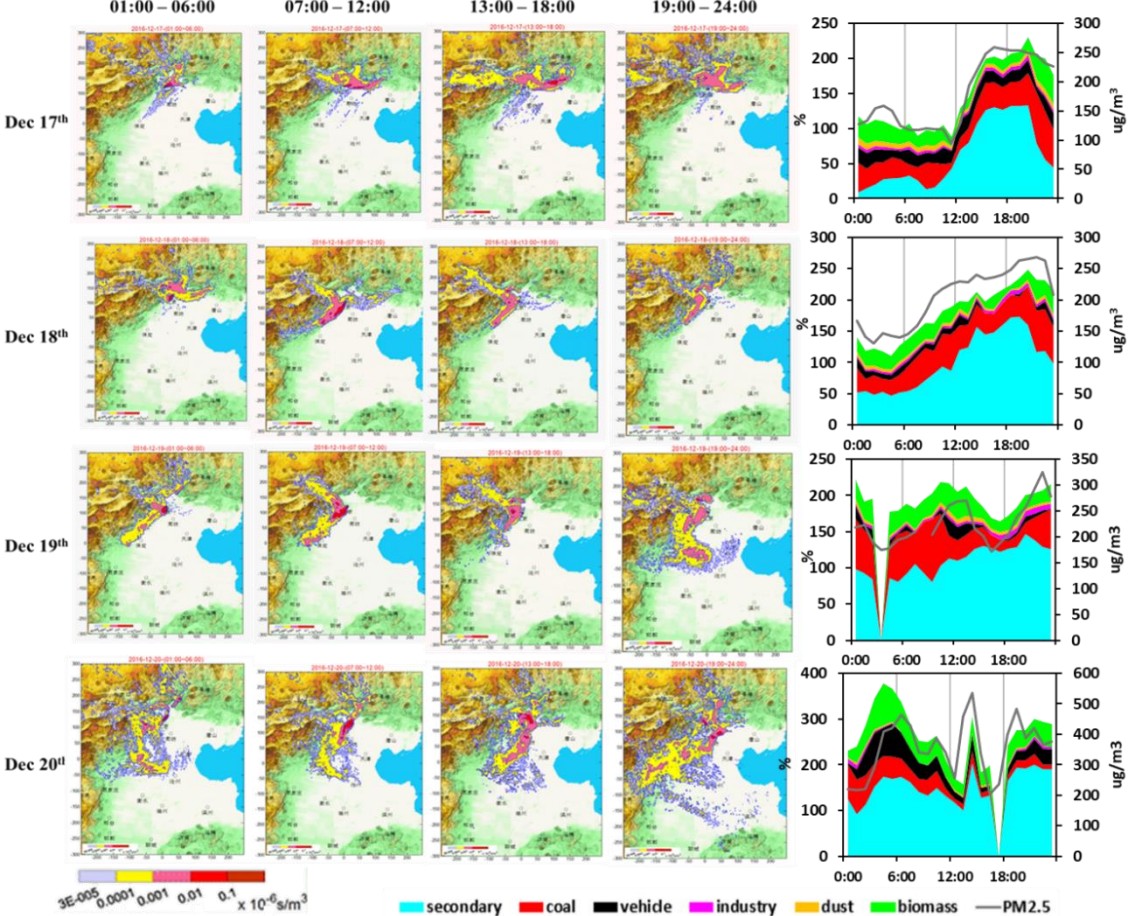

**Figure 7**. Source regions by the footprint model (every 6 h) and daily source apportionment results by PMF in EP4.






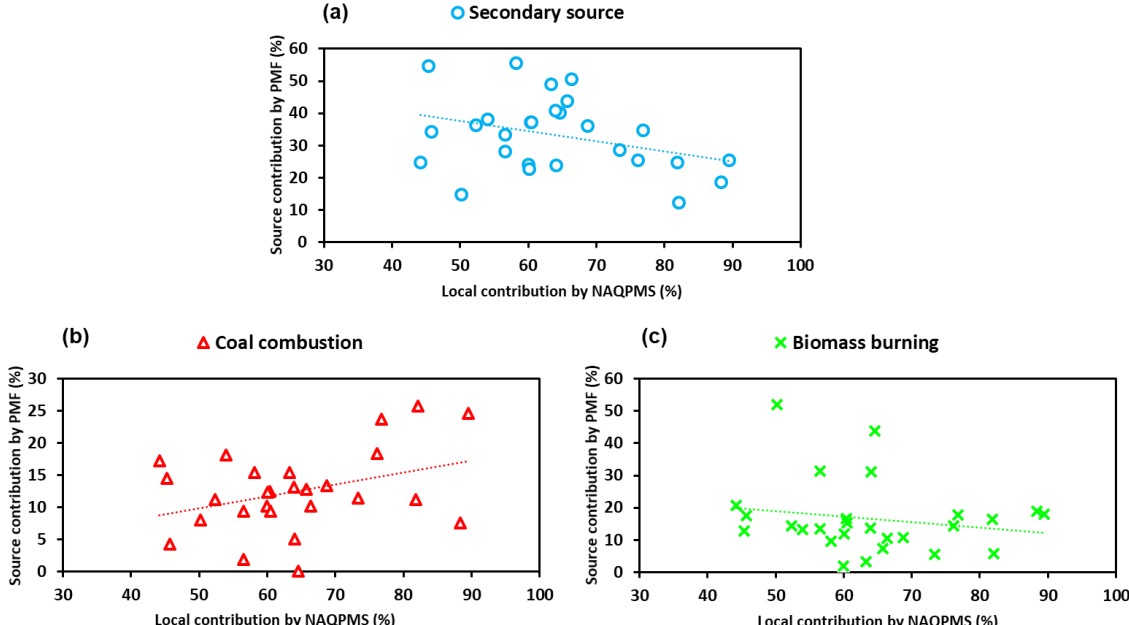

**Figure 8.** Correlations of local contribution by NAQPMS with the relative contribution by PMF of (a) secondary source, (b) coal combustion source and (c) biomass burning source.






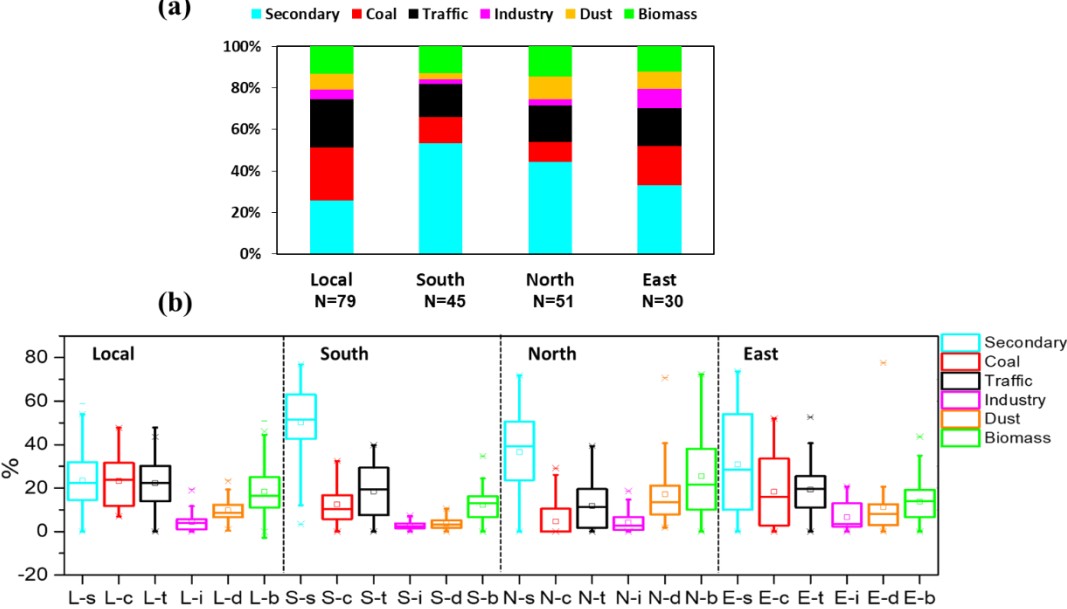

**Figure 9.** (a) The average source contribution (in percentage) for each type of footprint, and (b) box chart of source contribution in four types of footprint during the whole sampling period. N in (a) represents for the number of cases. The capital letters in (b) stands for the type of footprint (L for local; S for south; N for north; E for east) and the lowercases stands for different sources (s for secondary source, c for coal combustion, t for traffic source, i for industrial source, d for dust, and b for biomass burning)