# Peer review of "High-time Resolution Source Apportionment of PM2.5 in Beijing with Multiple Models"

_Atmospheric Chemistry and Physics, 2018_

## Referee Comment (RC1) · Anonymous Referee #1 · 20 Dec 2018

This study discussed source apportionment of PM2.5 in Beijing based on multiple models including PMF, footprint and NAQPMS models. However, there are too many studies with similar topics to resolve the sources of fine particle in Beijing. Nothing new findings and scientific questions were summarized in the abstract and conclusions even though it based on the high-time resolution data. Moreover, there are several questions and comments as follows needed to be addressed to improve the quality of this manuscript. 1. I suggest the authors to reconstruct the hourly PM2.5 mass based on the online chemical components (e.g. ions, OC/EC, and elements). The convert factor of OC to OM should be related to the sources rather than 1.8 which obtained in the regional parks in USA. Thus, the relationship between OC and EC should be discussed firstly. 2. I suggest the authors to present the data quality assurance and

control in more detail in section 2.2. For example, will the reconstructed PM2.5 match the measured PM2.5? What are the relationships between the elements (such as S, K, Cl and Ca) measured by Xact 625 and their water soluble form (SO42-, K+, Cl- and Ca2+) measured by IGAC. Moreover, the EC concentrations obtained by Sunset are also suggested to compare with the BC concentrations measured by MAAP or AE-31 etc. There are quite important for the source apportionment. 3. I suggest the authors to summarize the mass concentrations of chemical components in Table rather than in Figure for different episodes. It is important to judge the online data quality when compared to the previous studies in Beijing. 4. I don't think the explanation for the six factors in section 3.2 is reasonable, and it is not suitable for PMF to resolve PM sources with too many episode cases. The uncertainties of source contributions would be huge. For example, the contributions of the dominant sources during the episode cases would be overestimated during the non- episode period. 5. I also don't think the identified source profiles are reasonable. Firstly, K can be also originated from dust. Why not use K+? More fraction of K was showed in the industrial source rather than biomass burning. Secondary, industrial sources should be explained in detail. Why only little fractions of OC and EC in this source profile? Thirdly, why large fraction of Cl- was found in traffic emission? Lastly, why large fraction of Ni was found in biomass burning source? 6. What are the relationships between the tracers of identified sources and sources mass concentrations? 7. I don't think the discussions in 3.3 and 3.4 are necessary if the authors can't reply the above comments. 8. I suggest the authors to add more information about the spatial mass concentrations of PM2.5, PM10, SO2 and NO2 in Figure 7. Moreover, sources inventories used in this study would be suggested to add in the supplementary materials. The results resolved by the footprint and NAQPMS models should be discussed based on above information.

---

## Referee Comment (RC2) · Anonymous Referee #2 · 2 Jan 2019

General comment: The manuscript examines the influence of combining multiple models to better estimate the sources and regions of pollution in Beijing during winter, 2016. This is achieved by introducing the ambient concentrations of carbonaceous species (OC and EC), inorganic ions and selected metals to PMF model to apportion PM2.5; along with the foot print analysis and NAQPMS model. While PMF was the key model to apportion PM2.5 sources, further details about the optimum solution of PMF need to be discussed systematically, such as; the examination of the optimum factor solution, factor analysis, and the uncertainties associated with the estimation of each factor. I think it is important to add these details to the supplement and refer to them in the main text, as needed. Also, many of your comparisons with the previous study need to include more details, such as size fraction of PM, type of receptor model used and

weather organic tracers were used or not, time resolution, which month, year, etc. Beside these two major comments, I recommend accepting this manuscript for publication after corrections as detailed in the following.

Technical comments: 1. Page 1, line 25: Subscript PM2.5 to PM2.5

2. Page 1, line 26: Define "receptor model" and replace the name with the new sentence in the same line

3. Page 2, line 39: Why you have chosen to report 11% of the dust contribution only? I recommend adding the percent contribution for the major sources or each analyzed footprint, systematically.

4. Page 2, line 48: Insert a citation after "….life and human health".

5. Page 2, line 51, Coal combustion can also considered as industrial source. Please be more specific here.

6. Page 2, line 51: It sounds like there are many sources related to traffic. I suggest replace "traffic sources" with "traffic-related sources", after visiting the two papers you have cited here.

7. Page 3, line 62: "The model can…". Which model you are referring to? Please specify.

8. Page 3, line 77: Insert citation after "…in previous studies".

9. Page 5, line 106: "The room". Which room you are referring to?

10. Page 6, line 130: Here you report that XRF was used to quantify metals. I see that you need to add an excel sheet or a table to the supplement that shows: measured concentrations, uncertainties of the measurements, and the detection limit. Also, it should include PM2.5, EC, OC, SIA. These data are important for the science community to replicate the PMF result. Also, in many places later you report the averages of a certain species without the standard deviation and/or the range of that average, which

can also be extracted from the suggested table.

11. Page 8, line 181: Add a comma after "5 km".

12. Page 8, line 181: Add a space after "x", and before "2.5".

13. Page 10, line 229-232: you have calculated the concentration of mineral species (Al, Si, Fe) based on Ca concentration, and the composition of urban soil. Dose this typical urban soil was affected by regional and local pollution? During summer or winter? During hazy or non-hazy effect? And what is the estimated uncertainty in this calculation (estimation).

14. Page 10, line 230-232: Belong to the method section. Please move them.

15. Page 10, line 234: You have stated that Al and Si might be underestimated. Why? And by how much? Please provide supporting details.

16. Page 10, line 241: Here you compare the average OC/EC ratio with Yan et al., 2015. Can you be more specific about the time resolution, duration, months, and/or any special pollution events.

17. Page 10, line 242-243: Is the concentrations of SO42- (23±20) ug/m3 significantly higher than that for NO3- (22 ± 23)ug/m3? For me they look the same, taking the high variation of the concentrations, as they are reported. Please check these comparisons here and in other places along the manuscript. Also, discuss what is the potential reason for this observation based on previous PM2.5 studies conducted in Beijing during winter.

18. Page 10, line 246: Insert a citation(s) after "...in winter in Beijing"

19. Page 10, line 246-247: What type of combustion source are usually attributed to K and SO4, please be more specific and include a citation.

20. Page 10, line 249-250: " ...the contribution of sulfate increased significantly (up to 24%), compared to what? Please be more specific here. I think you want to say that

sulfate is the major component when PM2.5< 50ug/m3.

21. Page 10, line 249-250: You are using SO42- and sulfate back and forth. Choose one term and stick with it.

22. Page 11, line 254-255: Insert citations after "…tracers of biomass burning and coal combustion", and after "tracers of dust sources".

23. Page 11, line 270: Add a "comma" after Ba.

24. Page 12, line 277-278: Add an estimation for the of cooking sources to PM2.5 in Beijing during winter, based on studies utilized organic tracers during winter. And discuss weather it is a minor or major source.

25. Page 12, line 287: Add a SD for the average value of 75 ug/m3. Or you can report the average PM2.5 during the four haze episodes > 98ug/m3 to be consistent with the table.

26. Page 14, line 348-349: Here you almost restated the previous paragraph (line341-345). I think it is not necessary.

27. Page 15, line 271: We can control precursors of secondary sources, but not the secondary sources. Please modify accordingly.

28. Page 20, line 523: Fix (PM2. 5). Extra space

29. Page 20, line 532: Capitalize the first word of the title only.

30. Page 22, line 569: Sulfate and nitrate (check technical comment #21).

31. Page 24, Figure two: I suggest naming them a and b. Also, please explain the what the white bars represent?

32. Page 26, Figure 4: The right side of the Y-axis shows more than 100%. These are % of what?

33. Page 27, Figure 5: Move the boxes of PMF source identifiers to the left side of

the figure and locate them under source apportionment results only. Also, it would be better if you rename these figures as a, and b.

34. Page 28, Figure 6: same comment as for (technical comment #32).

35. Page 29, Figure 7: Check technical comment # 33.

36. Page 30, Figure 30: Add r and p value for the correlations. And discuss in the text.
* * *

---

## Referee Comment (RC3) · Anonymous Referee #3 · 10 Jan 2019

In this study, the authors implemented a combination of PMF, footprint, and NAQPMS models to identify the sources and regions of PM2.5 in Beijing. The PMF model was used to find out the sources of PM2.5. To this end, the chemical components of PM2.5 were used as the input to the PMF model. Then the PMF results were combined with footprint and NAQPMS models to identify the evolution of different episodes. Overall, the paper is well written in English and the results are promising. However there are some comments which need to be addressed in order to make it a good candidate for publication in ACP. I believe that the paper can be accepted after addressing these comments:

Line 52- please add two or three more references for the PM2.5 source apportionment studies. For example you might add the following papers:

Kotchenruther, R. a., 2016. Source apportionment of PM2.5 at multiple Northwest U.S. sites: Assessing regional winter wood smoke impacts from residential wood combustion. Atmos. Environ. 142, 210–219.

Taghvaee, S., Sowlat, M.H., Mousavi, A., Hassanvand, M.S., Masud, Y., Naddafi, K., Sioutas, C., 2018. Source apportionment of ambient PM 2.5 in two locations in central Tehran using the Positive Matrix Factorization ( PMF ) model. Sci. Total Environ. 629,

Zong, Z., Wang, X., Tian, C., Chen, Y., Qu, L., Ji, L., Zhi, G., Li, J., Zhang, G., 2016. Source apportionment of PM2.5 at a regional background site in North China using PMF linked with radiocarbon analysis: Insight into the contribution of biomass burning. Atmos. Chem. Phys. 16, 11249–11265.

Line 106- I suggest you to add references for your claim that "the sampling site is representative of the Beijing urban area" (if applicable).

Line 111- You need to mention more details regarding the usage of Semi-continuous OC/EC Carbon Aerosol Analyzer (Sunset Laboratory Inc.)  (e.g., thermal protocols used). Please also add references for the instrument.

Line 142- You definitely need to present the average concentration of PM2.5 chemical components in a table for different episodes of your study.  This table should also include the min, max, signal/ noise (S/N) ratio for your data as the important parameters in PMF analysis.

Line 149- please add the (Norris et al., 2014; Paatero and Tapper, 1994; Paatero et al., 2014;Paatero, 1997) as the main references for PMF model:

Paatero, P., 1997. Least Squares Formulation of Robust Non-negative Factor Analysis. pp. 23–35.

Paatero, P., Tapper, U., 1994. Positive matrix factorization: a non-negative factor model with optimal utilization of error estimates of data values. Environmetrics 5, 111–126.

Paatero, P., Eberly, S., Brown, S.G., Norris, G.a., 2014. Methods for estimating uncertainty in factor analytic solutions. Atmos.Meas. Tech. 7:781–797. https://doi.org/10.5194/amt-7-781-2014.

Norris, G., Duvall, R., Brown, S., Bai, S., 2014. EPA Positive Matrix Factorization (PMF) 5.0 Fundamentals and User Guide.

Line 163- Please provide the Q robust values for different PMF solutions in an SI figure. This would be really helpful in showing why you picked the 6 factor solution as the optimal PMF resolved solution.

Line 166- In addition to briefly touching the results of your uncertainty analysis, you need to mention the uncertainty analysis results in detail (more discussions can be found in PMF source apportionment papers)

—Why the simulation period for footprint model, and NAQPMS model are not the same? For example, the footprint simulation was performed from 1-31 December while the NAQPMS model analysis was performed from 10th of November to 15th of December.

Line 247- Please add a couple of references for the following sentence:

In general, the large contribution of SIA, OM as well as the high OC/EC ratio indicated the importance of secondary formation in winter in Beijing, while the high concentration of species like $SO_4^{2-}$ and K suggested a significant contribution of combustion sources to $PM_{2.5}$.

Line 255- As a general comment, you need to add references while mentioning different chemical components as tracers of a specific source. For example, references are required for the fact that K is a tracer of biomass burning.

Line 260-275: Unfortunately, the source apportionment profiles are not distinguished well. For example, K as a tracer of biomass burning has higher percentage of contribution in Industrial sources rather than the biomass burning. In addition, we have significant loadings of Na+ and Ni (which are not tracers of biomass burning) in biomass burning profile. How do you justify your source profiles?

Line 335- How do you compensate the lack of data for regional and local contribution from the NAQPMS model for the EP4?

Line 425- Authors should include the limitations of their research. Please add the limitations as a separate session.

---

## Author Response (AR1)

**Responses to Comments by Reviewer #1, Reviewer #2 and Reviewer #3**

**Ms. Ref. No.:** acp-2018-1234
**Title:** High Time Resolution Source Apportionment of PM$_{2.5}$ in Beijing with Multiple Models
**Authors:** Yue Liu et al.

**Corresponding author:** Mei Zheng (email: mzheng@pku.edu.cn)

We are grateful for the helpful comments by all the reviewers. Based on these comments, we have carefully revised the manuscript. The response to each comment is listed below. The original comments from reviewers are in *blue and italic*, our replies are in normal font and the tracked changes in the revised manuscript are in *red and italic*. The page and line number refers to the revised manuscript without marks.

**Reviewer #1 Comment No.1**: *I suggest the authors to reconstruct the hourly PM2.5 mass based on the online chemical components (e.g. ions, OC/EC, and elements). The convert factor of OC to OM should be related to the sources rather than 1.8 which obtained in the regional parks in USA. Thus, the relationship between OC and EC should be discussed firstly.*

**Response to reviewer comment NO.1:** We agree with the reviewer that the convert factor of OC to OM ($R_{oc}$) should be related to the sources. According to previous studies, $R_{oc}$ would be relatively higher for samples influenced by biomass burning event (Malm et al., 2007; Poirot and Husar., 2004; Turpin and Lim., 2001). For example, Turpin and Lim (2001) have reported $R_{oc}$ values ranging from 2.2 to 2.6 for samples with impacts from biomass burning. Poirot and Husar (2004) found good agreement between measured and reconstructed fine mass by applying a $R_{oc}$ factor of 1.8 during a biomass burning event. Besides, $R_{oc}$ of urban aerosol tends to be lower than that of nonurban aerosol because nonurban areas are likely to have higher contributions of both biogenic and secondary anthropogenic sources than observed in urban areas (Turpin and Lim., 2001). Turpin and Lim (2001) recommend a factor of 1.6±0.2 for urban organic aerosols, a factor of 2.1±0.2 for nonurban organic aerosols. The $R_{oc}$ of 1.6 has also been used in studies of urban areas in China (Cao et al., 2007; Huang et al., 2017). According to the source apportionment results in our study, the contribution of biomass burning source was not very significant during the sampling period (9% on average, see Figure 3 in the manuscript), and the sampling site is more influenced by the urban sources of Beijing. Therefore, we change the $R_{oc}$ from 1.8 to 1.6 which is more

fit for our case. The related description has been revised as follows (see Page 6, Line 151):

*Organic matter (OM) was calculated as OM= 1.6 ×OC (Turpin and Lim, 2001).*

After the revision of $R_{oc}$, we reconstructed the hourly $PM_{2.5}$ mass based on the online chemical components (see Figure 1). The correlation between the measured and reconstructed $PM_{2.5}$ is good with the slope close to 1.0 and the $R^2$ over than 0.9 (p< 0.05, n=1099). Figure 1 has also been added in the supplementary materials as **Figure S6**.

[Figure]

Figure 1 Correlation of measured and reconstructed $PM_{2.5}$ mass

**Reference**

[1]  Cao, J. J., Lee, S. C., Ho, K. F., Zou, S. C., Fung, K., Li, Y., Watson, J. G., and Chow, J. C.: Spatial and seasonal variations of atmospheric organic carbon and elemental carbon in Pearl River Delta Region, China, Atmos. Environ., 38, 4447-4456, 2004.

[2]  Huang, X., Liu, Z., Liu, J., Bo, H., and Wang, Y.: Chemical characterization and source identification of $PM_{2.5}$ at multiple sites in the Beijing-Tianjin-Hebei region, China, Atmos. Chem. Phys., 17, 12941-12962, 2017.

[3]  Lim, H.J., and Turpin, B. J.: Origins of primary and secondary organic aerosol in Atlanta: Results of time-resolved measurements during the Atlanta supersite experiment, Environ. Sci. Technol., 36, 4489-4496, 2002.

[4]  Malm, W. C., and Hand, J. L.: An examination of the physical and optical properties of aerosols collected in the IMPROVE program, Atmos. Environ., 41, 3407-3427, 2007.

[5]  Poirot, R.L., and Husar, R.B.: Chemical and physical characteristics of wood smoke in the northeastern US during July

2002: impacts from Quebec forest fires. A&WMA Specialty Conference: Regional and Global Perspectives on Haze: Causes, Consequences and Controversies, Asheville, NC, 2004.

**Reviewer #1 Comment No.2**: *I suggest the authors to present the data quality assurance and control in more detail in section 2.2. For example, will the reconstructed PM2.5 match the measured PM2.5? What are the relationships between the elements (such as S, K, Cl and Ca) measured by Xact 625 and their water soluble form (SO42-, K+, Cl- and Ca2+) measured by IGAC. Moreover, the EC concentrations obtained by Sunset are also suggested to compare with the BC concentrations measured by MAAP or AE-31 etc. There are quite important for the source apportionment.*

**Response to reviewer comment NO.2:** According to *Response to reviewer comment NO.1*, the reconstructed and the measured $PM_{2.5}$ mass agree well with the slope of 1.0 and the $R^2$ over than 0.9 (see Figure 1). About quality control, we have added more detailed information in the supplementary materials including the concentration of the internal standard (Pd) of Xact and the internal standard (LiBr) of IGAC. As shown in Figure 2 (**Figure S1** in the supplementary materials), for IGAC the internal standard (LiBr) was added continuously to each sample and analyzed by the IC system during the analysis to check the stability of the IGAC instrument. During the sampling period, the mean concentrations of $Li^+$ and $Br^-$ were within the range of three standard deviations, suggesting a stable condition of the IGAC. As shown in Figure 3 (**Figure S3** in the supplementary materials), for the Xact a Pd rod was used as automatic internal quality control to check the performance of the instrument on a daily basis and the mean concentration of Pb was within the range of three standard deviations during the sampling period.

We have collected offline samples simultaneously with online measurement at the same site and compared the online concentration of ion species measured by IGAC with those measured by offline ion chromatography (Zhang et al., under review). The results are shown in Figure 4. The correlation of $SO_4^{2-}$, $NO_3^-$, $Cl^-$ and $Na^+$ between online and offline measurement is relatively good ($R^2 > 0.8$) while the correlation of $Ca^{2+}$, $K^+$ and $Mg^{2+}$ is relatively poor. We also compared the online concentration of elemental species measured by Xact with those measured by offline ICP-MS and found better correlation ($R^2 > 0.9$) for most species such as K, Cr, Mn, Fe, Cu, Ni, Zn, As and Pb (Zhang et al., under review). Therefore, in our study, after quality check, we use 12 species measured by Xact while only use the $SO_4^{2-}$, $NO_3^-$, $Cl^-$ and $Na^+$ concentration measured by IGAC for further discussion. Based on the comparison between online and offline measurements, K and Ca show better correlation than $K^+$ and $Ca^{2+}$, therefore,

only K and Ca data by Xact are used in the analysis in this study.

During the sampling period, we also measured BC by AE33 at the same site but we did not use MAAP. However, we used a Sunset OC/EC analyzer at the same time. The correlation of BC measured by AE33 and EC measured by Sunset OC/EC analyzer is shown in Figure 5. Relatively high correlation could be found between the two instruments with the slope of 1.0 and the $R^2$ over than 0.9.

[Figure]

Figure 2. Concentration of the internal standard (LiBr) of IGAC

[Figure]

Figure 3. Concentration of the internal standard (Pd) of Xact

[Figure]

Figure 4. Correlation between online IGAC and offline method for ion species (Zhang et al., under review)

[Figure]

Figure 5. Correlation of BC measured by AE33 and EC measured by Sunset OC/EC analyzer

**Reference**

[1]   Zhang, B. Y., Tian, Z., Yan, C. Q., Li, X. Y., Yu, J. T., Wang, S. X., Liu, B. X., Zheng, M.: Comparison of water-soluble inorganic ions and trace metals in PM$_{2.5}$ between online and offline measurements in Beijing during winter, under review.

**Reviewer #1 Comment NO.3**: *I suggest the authors to summarize the mass concentrations of chemical components in Table rather than in Figure for different episodes. It is important to judge the online data quality when compared to the previous studies in Beijing.*

**Response to reviewer comment NO.3:** According to the reviewer's suggestions, the mass concentration of chemical components has been summarized in Table 1 and Table 2 as follows, and also added in the supplementary material as Table S1 and Table S4:

**Table 1 Statistical summary of identified species of PM$_{2.5}$ in the entire sampling period**

n=1099

|  | Mean | Std. | Max | Min | Detection limit | BDL% |
|---|---|---|---|---|---|---|
| **OC/EC** |  |  | **µg/m$^3$** |  |  | **%** |
| OC | 20.8 | 17.0 | 89.9 | 1.1 | 0.4 | - |
| EC | 5.6 | 4.4 | 23.1 | 0.2 | 0.1 | 4.3 |
| **SIA** |  |  | **µg/m$^3$** |  |  | **%** |
| SO$_4^{2-}$ | 23.5 | 20.8 | 95.8 | 0.04 | 0.04 | 0.21 |
| NO$_3^-$ | 22.0 | 23.3 | 104.7 | 0.03 | 0.03 | 0.1 |
| NH$_4^+$ | 14.0 | 14.7 | 66.6 | 0.04 | 0.05 | 1.4 |
| Na$^+$ | 0.39 | 0.32 | 1.89 | 0.02 | 0.04 | 8.5 |
| Cl$^-$ | 4.89 | 4.19 | 27.6 | 0.05 | 0.05 | 0.1 |
| **Metal** |  | **µg/m$^3$** |  |  | **ng/m$^3$** | **%** |
| K | 1.49 | 1.17 | 5.28 | 0.10 | 2.366 | - |
| Fe | 0.769 | 0.541 | 2.22 | 0.015 | 0.759 | - |
| Ca | 0.384 | 0.277 | 2.08 | 0.001 | 0.902 | - |
| Zn | 0.286 | 0.261 | 1.85 | 0.005 | 0.231 | - |
| Pb | 0.107 | 0.091 | 0.469 | 0.004 | 0.218 | - |

| | | | | | | |
|---|---|---|---|---|---|---|
| Mn | 0.058 | 0.046 | 0.210 | 0.001 | 0.283 | - |
| Ba | 0.035 | 0.023 | 0.160 | 0.002 | 0.945 | - |
| Cu | 0.027 | 0.024 | 0.171 | 0.002 | 0.267 | - |
| As | 0.022 | 0.021 | 0.084 | 0.000 | 0.114 | 17 |
| Cr | 0.010 | 0.010 | 0.110 | 0.000 | 0.288 | 11 |
| Se | 0.008 | 0.008 | 0.046 | 0.000 | 0.141 | 5.2 |
| Ni | 0.002 | 0.002 | 0.044 | 0.000 | 0.226 | 3.4 |

* BDL% refers to the percentage of data below the detection limit

* The unit of the detection limit of each metal is $ng/m^3$

* '-' means that all data are above the detection limit

**Table 2 Average concentration of $PM_{2.5}$ and identified species in different haze and non-haze periods**

Unit: $\mu g\ m^{-3}$

| Average Conc. | EP1 n=102 | EP2 n=95 | EP3 n=117 | EP4 n=131 | NH1 n=78 |
|---|---|---|---|---|---|
| $PM_{2.5}$ | 97.7±70.7 | 143.8±119.1 | 115.3±108.6 | 241.8±115.5 | 18.8±20.7 |
| OC | 19.1±10.7 | 24.7±18.7 | 23.1±21.1 | 40.3±14.5 | 3.33±2.85 |
| EC | 4.1±2.7 | 7.1±5.1 | 5.6±4.3 | 11.0±3.7 | 0.9±0.6 |
| $SO_4^{2-}$ | 18.6±10.9 | 25.1±20.4 | 23.4±19.7 | 53.3±19.2 | 4.43±3.89 |
| $NO_3^-$ | 19.9±14.6 | 18.9±16.0 | 23.3±23.3 | 56.8±24.8 | 2.81±3.69 |
| $NH_4^+$ | 13.3±8.36 | 12.6±11.6 | 13.3±13.2 | 36.4±16.7 | 2.42±3.80 |
| $Na^+$ | 0.37±0.23 | 0.35±0.28 | 0.49±0.41 | 0.69±0.23 | 0.10±0.14 |
| $Cl^-$ | 5.07±3.05 | 5.17±3.71 | 5.66±5.87 | 8.15±3.41 | 0.97±1.10 |
| K | 2.55±1.33 | 1.21±0.90 | 1.23±1.08 | 2.53±0.78 | 0.410±0.410 |
| Fe | 1.26±0.58 | 0.724±0.568 | 0.641±0.480 | 1.06±0.25 | 0.144±0.133 |
| Ca | 0.530±0.250 | 0.479±0.447 | 0.356±0.249 | 0.360±0.148 | 0.054±0.033 |
| Zn | 0.442±0.242 | 0.242±0.197 | 0.252±0.289 | 0.409±0.170 | 0.060±0.076 |
| Pb | 0.182±0.105 | 0.087±0.071 | 0.090±0.084 | 0.185±0.063 | 0.024±0.028 |
| Mn | 0.093±0.050 | 0.049±0.038 | 0.049±0.045 | 0.082±0.021 | 0.011±0.012 |
| Ba | 0.047±0.016 | 0.038±0.032 | 0.032±0.022 | 0.048±0.012 | 0.006±0.005 |

| | | | | | |
|---|---|---|---|---|---|
| Cu | 0.028±0.015 | 0.023±0.021 | 0.028±0.026 | 0.042±0.017 | 0.007±0.009 |
| As | 0.031±0.019 | 0.019±0.019 | 0.021±0.023 | 0.040±0.013 | 0.003±0.006 |
| Cr | 0.019±0.022 | 0.007±0.009 | 0.008±0.011 | 0.014±0.010 | 0.001±0.001 |
| Se | 0.012±0.007 | 0.006±0.005 | 0.007±0.007 | 0.018±0.007 | 0.001±0.001 |
| Ni | 0.003±0.002 | 0.002±0.001 | 0.002±0.002 | 0.004±0.004 | 0.0006±0.0004 |

**Reviewer #1 Comment NO.4**: *I don't think the explanation for the six factors in section 3.2 is reasonable, and it is not suitable for PMF to resolve PM sources with too many episode cases. The uncertainties of source contributions would be huge. For example, the contributions of the dominant sources during the episode cases would be overestimated during the non- episode period.*

**Response to reviewer comment NO.4:** This is a very helpful comment. We think that the reviewer may concern that the major source types might change from haze episodes to non-haze periods and we agree that the uncertainties of PMF would be big if the main source types change during the sampling period. However, the sampling period of our study was within the heating period in winter. In Beijing, source types would change with season (e.g., more dust in spring), but within the same season, it does not show a significant change (Lv et al., 2016).

We agree that this is an important concern. To test this, we separate the data in haze episodes and non-haze periods into two different input files and apply PMF for source apportionment separately. The result based on haze episode data is represented by EP-PMF, the result based on non-haze period data is represented by NH-PMF and the previous PMF result based on all the input data is showed as PRE-PMF. The average source contributions in EP1, EP4 and NH1, NH2 calculated by EP-PMF, NH-PMF and PRE-PMF are shown in Figure 6. It could be seen that the source contributions in different haze episodes and non-haze periods did not change significantly based on different input of PMF.

[Figure]

Figure 6. The average source contributions in EP1, EP4 and NH1, NH2 calculated by EP-PMF, NH-PMF and PRE-PMF

**Reference**

[1]   Lv, B., Zhang, B., and Bai, Y.: A systematic analysis of $PM_{2.5}$ in Beijing and its sources from 2000 to 2012, Atmos. Environ., 124, 98-108, 2016.

**Reviewer #1 Comment NO.5**: *I also don't think the identified source profiles are reasonable. Firstly, K can be also originated from dust. Why not use K+? More fraction of K was showed in the industrial source rather than biomass burning. Secondary, industrial sources should be explained in detail. Why only little fractions of OC and EC in this source profile? Thirdly, why large fraction of Cl- was found in traffic emission? Lastly, why large fraction of Ni was found in biomass burning source?*

**Response to reviewer comment NO.5:** According to *Response to reviewer comment NO.2*, only the concentration of $SO_4^{2-}$, $NO_3^-$, $Cl^-$ and $Na^+$ measured by IGAC were used for further discussion in the manuscript after performing comparison between online and offline measurement data. We agree with the reviewer that $K^+$ is a better indicator for biomass burning than K. However, besides poor agreement

between online and offline data for $K^+$, a certain fraction (about 25%) of $K^+$ measured by IGAC was below or close to its detection limit (0.05 μg/m$^3$) during the sampling period. Therefore, $K^+$ was not included in the input of PMF and K was used instead. There have been studies in which K was used as a tracer for biomass burning source when $K^+$ was not available (Song et al., 2006; Yu et al., 2013; Zheng et al., 2014; Gao et al., 2016; Z kov á et al., 2016). The correlation of $K^+$ concentration with the biomass burning source identified by PMF model are shown in Figure 7. It could be seen that $K^+$ exhibited a relatively good correlation with the contribution of biomass burning source from PMF with K as input data, with $R^2$ over 0.6.

In our PMF results, industrial source and biomass burning source were the two major sources of K in Beijing, China. This conclusion could be supported by the local source profiles in China shown in Figure 8 (Dai et al., 2015; Cao et al., 2015; Ma et al., 2015; Wang et al., 2009; Wen et al., 2009; Wu et al., 2016; Zhang et al., 2012; Zheng et al., 2013). As shown in Figure 8(c), the average fraction of K in industrial source profiles (including coal-fired power plant and boiler, steel and iron plant in Beijing, Shanghai and Jilin) was similar with that in biomass burning source profiles (including wheat, corn and rice), which is about 8%. Moreover, in some industrial source profiles the fraction of K exceeded that in biomass burning source profiles. The potential reasons of K in industrial source including that coal was used in some power plants and the use of potassium feldspar in glass and fertilizer industry (Ma et al., 2010). Therefore, this might be the reason that K is found in industrial source identified by the PMF analysis. As shown in Figure 8(a) and (b), the fraction of OC and EC in industrial profiles were much lower than those in coal combustion source and biomass burning source, which might be attributed to the sufficient pretreatment process and higher combustion efficiency in industries (Wang et al., 2009; Zheng et al., 2013). Therefore, lower OC and EC are seen in industry factor of our PMF results. In secondary source profile, EC is low because it is mostly emitted from primary sources including traffic source. OC is relatively higher than EC due to the formation of secondary organic carbon.

For the third question, we agree with the reviewer that Cl$^-$ has not been used as a specific tracer for traffic source in previous studies. In our study, large fraction of Cl$^-$ was observed in the profiles of coal combustion and biomass burning source which corresponded well with previous studies (Frigge et al., 2016; Lobert et al., 1999). However, as shown in Figure 9, we also have found relatively high fraction of Cl$^-$ in traffic source profiles in some previous studies. Except for the predominant species OC and EC, the fraction of Cl$^-$ is relatively high compared to other species, which might be attributed to the emission of rubber accessory (neoprene) of vehicles and the use of chlorine in the catalytic reforming of gasoline (Chen et al., 2015; Chow et al., 2004; Cui et al., 2016; Watson et al., 2001; Schauer et al., 1999; Veksha

et al., 2018). Similar with Cl⁻ in traffic source, Ni in biomass burning source has not been regarded as a specific tracer. However, during the process of biomass gasfication, Ni is commonly used as a catalyst (Corella et al., 1999; Sutton et al., 2001).

In conclusion, our PMF results are based on the commonly used tracers and are supported by previous source profiles. It is not uncommon to find that the factor resolved by PMF could not be exactly the same as the source profile from emission source testing. In the future, with more samples and more tracers such as organic markers, it will help to achieve a quite distinct factor and very close to specific emission source profile.

[Figure]

Figure 7. Time series and correlation of K⁺ with the biomass burning source identified by PMF

[Figure]

Figure 8. Fraction of (a) OC, (b) EC and (c) K in different source profiles (Dai et al., 2015; Cao et al., 2015; Ma et al., 2015; Wang et al., 2009; Wang et al., 2016; Wen et al., 2009; Wu et al., 2016; Zhang et al., 2012; Zheng et al., 2013)

[Figure]

Figure 9. Source profiles of traffic sources in different country (USA1: Watson et al., 2001; USA2:

California, Schauer et al., 1999; USA3: Colorado, Watson et al., 2001; USA4: Texas, Chow et al., 2004; Mexico: Watson et al., 2001; China1: Nanjing, Chen et al., 2015; China2: Pearl River Delta, Feng et al., 2013; China3: Shandong-gasoline, Cui et al., 2016; China4: Shandong-diesel, Cui et al., 2016; China5: Shandong-truck, Cui et al., 2016)

**Reviewer #1 Comment NO.6**: *What are the relationships between the tracers of identified sources and sources mass concentrations?*

**Response to reviewer comment NO.6:** According to the reviewer's suggestions, we have added the relationships between the tracers of identified sources and sources mass concentrations in the supplementary materials as **Figure S8**. SO$_4^{2-}$ and NO$_3^-$ were typical of the secondary source profiles (Gao et al., 2016; Peng et al., 2016). Mn and Zn were used as indicators for industrial source (Hu et al., 2015; Li et al., 2017). Ca and Ba were mainly emitted from dust source (Amato et al., 2013; Shen et al., 2016). The correlation of EC and NOx was analyzed with traffic source and As and Se were typical tracers for coal combustion (Vejahati et al., 2010).

Figure 10(a1), (b1), (c1), (d1) and (e1) shows the time series of source concentration and source tracers of secondary source, coal combustion source, industrial source, traffic source and dust source. Figure 10(a2), (a3) shows the scatterplot of secondary source with $SO_4^{2-}$ and $NO_3^-$. Figure 10(b2), (b3) shows the scatterplot of coal combustion source with As and Se. Figure 10(c2), (c3) shows the scatterplot of industrial source with Mn and Zn. Figure 10(d2), (d3) shows the scatterplot of traffic source with EC and NOx. Figure 10(e2), (e3) shows the scatterplot of dust source with Ca and Ba. It could be seen that secondary source concentration showed high correlation with $SO_4^{2-}$ and $NO_3^-$ concentration, with both $R^2$ higher than 0.8. Coal combustion source concentration correlated well with As and Se concentration, with the $R^2$ of 0.86 and 0.98, respectively. Industrial source concentration correlated well with Mn and Zn, with $R^2$ over 0.85. As mentioned before, the correlation between biomass burning source and $K^+$ was good with $R^2$ over 0.6. For traffic source, we investigated its correlation with EC and NOx, and found relatively high correlation with $R^2$ of 0.70 and 0.51, respectively. Dust source concentration correlated well with the concentration of Ca and Ba, with $R^2$ of 0.96 and 0.72, respectively. In summary, the tracers of identified sources and sources mass concentrations show good correlation in general, which help to verify the PMF results. We appreciate this helpful suggestion by the reviewer.

[Figure]

Figure 10. Relationships between the tracers of identified sources and sources mass concentrations (secondary for secondary source; coal for coal combustion source; industry for industrial source; traffic for traffic source; dust for dust source). (a1)-(e1) shows the time series of source concentration with source tracers. (a2)-(e2) and (a3)-(e3) shows the scatterplot of source concentration with source tracers.

**Reviewer #1 Comment NO.7**: *I don't think the discussions in 3.3 and 3.4 are necessary if the authors can't reply the above comments.*

**Response to reviewer comment NO.7:** We have tried our best to reply the above comments and the detailed reply to each comment is listed above. Therefore, we would like to keep discussions in 3.3 and 3.4.

**Reviewer #1 Comment NO.8**: *I suggest the authors to add more information about the spatial mass concentrations of PM2.5, PM10, SO2 and NO2 in Figure 7. Moreover, sources inventories used in this study would be suggested to add in the supplementary materials. The results resolved by the footprint and*

*NAQPMS models should be discussed based on above information.*

**Response to reviewer comment NO.8:** Based on the reviewer's suggestion, the spatial mass concentrations of $PM_{2.5}$, wind speed and wind direction during EP1 in Figure 7 are added to the supplementary materials as **Figure S9** for better understanding of the evolution of the haze episode. The source inventory used in NAQPMS is the MIX (http://www.meicmodel.org/dataset-mix.html) anthropogenic emission inventory with the original resolution of 0.25 °(about 25 km at middle latitudes) and the year of 2010. It could be applied and downloaded on the website. The results of NAQPMS model are based on the MIX inventory while the footprint model do not include an inventory as its input.

**Reviewer #2 General Comment**: *While PMF was the key model to apportion PM2.5 sources, further details about the optimum solution of PMF need to be discussed systematically, such as; the examination of the optimum factor solution, factor analysis, and the uncertainties associated with the estimation of each factor.*

**Response to reviewer general comment:** Thanks for the reviewer's suggestion. We added the detailed information as to how to find the optimum factor solution and the uncertainty estimation by bootstrapping (BS), displacement (DISP), and bootstrapping with displacement (BS-DISP) in the manuscript and the supplementary materials.

  To determine the optimal number of source factors, a string of effective test, in which factors number was from four to nine, was carried out. The resulting Q parameters were shown in Figure 11 (**Figure S4** in the supplementary materials). Obviously, there was a lowest $Q_{Robust}$ value (13087) at six factors in moving from four to nine factors. Although $Q_{expected}$ has been decreasing in the process, $Q/Q_{expected}$ shared similar variation with $Q_{Robust}$ showing the lowest value at six factors (1.3).

  Uncertainty of PMF model is usually estimated by bootstrapping (BS), displacement (DISP), and bootstrapping with displacement (BS-DISP). Here, characteristics of factors nearby six, where $Q_{Robust}$ was relative lower, were explored. With five factors, three factors were mapped 100% of BS, while industry source and traffic source were mapped 92% and 94%, respectively of runs. There were no swaps with DISP, and 100% of the BS-DISP runs were successfully. At six factors, results were more stable with all factors mapped in BS in 100% (Table 3 below, also **Table S2** in the supplementary materials), no swaps occurred with DISP and all BS-DISP runs were successfully. However, the solution became less stable

in moving from six to seven factors. The new sea salt factor was only mapped in BS in 87% and coal combustion factor was mapped in BS in 89%, traffic source factor was mapped in 93%, other factors were mapped in 100% of runs. No swaps were found in DISP. Therefore, based on the above analysis, six factors were found to be the optimal solution in this study.

Table 3 Percentage of BS factors assigned to each base case factor with a correlation threshold of 0.6.

| Boot Factor | Secondary | Industrial | Dust | Traffic | Coal | Biomass |
|---|---|---|---|---|---|---|
| 1 | 100 | 0 | 0 | 0 | 0 | 0 |
| 2 | 0 | 100 | 0 | 0 | 0 | 0 |
| 3 | 0 | 0 | 100 | 0 | 0 | 0 |
| 4 | 0 | 0 | 0 | 100 | 0 | 0 |
| 5 | 0 | 0 | 0 | 0 | 100 | 0 |
| 6 | 0 | 0 | 0 | 0 | 0 | 100 |

[Figure]

Figure 11. The variation of Q parameters from four factors to ten factors

**Reviewer #2 General Comment**: *Also, many of your comparisons with the previous study need to include more details, such as size fraction of PM, type of receptor model used and weather organic tracers were used or not, time resolution, which month, year, etc.*

**Response to reviewer general comment:** According to the reviewer's suggestion, the detailed information including size fractions of previous studies has been added in the supplementary materials as follows:

Table 4. Previous studies about source apportionment of Beijing

|  | Sampling period and time resolution | Size fraction | Receptor model | Tracers |
|---|---|---|---|---|
| Gao et al., 2016 | July to August, 2014; 1 hour | 2.5 μm | PCA; PMF; ME2 | Inorganic tracers |
| Peng et al., 2016 | July to August, 2014; 1 hour | 2.5 μm | ME2 | Inorganic tracers |
| Zhang et al., 2013 | 2009-2010; daily | 2.5 μm | PMF | Inorganic tracers |

**Reviewer #2 Comment NO.9**: *Page 5, line 106: "The room". Which room you are referring to?*

**Response to reviewer comment NO.9:** "The room" refers to the sampling site and it has been revised as follows (see Page 5, line 108):

*The sampling site is located on the sixth floor of a teaching building within PKU.*

**Reviewer #2 Comment NO.10**: *Page 6, line 130: Here you report that XRF was used to quantify metals. I see that you need to add an excel sheet or a table to the supplement that shows: measured concentrations, uncertainties of the measurements, and the detection limit. Also, it should include PM2.5, EC, OC, SIA. These data are important for the science community to replicate the PMF result. Also, in many places later you report the averages of a certain species without the standard deviation and/or the range of that average, which can also be extracted from the suggested table.*

**Response to reviewer comment NO.10:** We agree with the reviewer about providing detailed information. Table 4 has been added in the supplementary materials as **Table S1** which shows the average concentration, standard deviation, detection limit and BLD% (the percentage of data below the detection limit) of each species, including OC, EC, SIA and metals. The table was presented as follows:

**Table 4 Statistical summary of identified species of $PM_{2.5}$ in the entire sampling period**

| | Mean | Std. | Max | Min | Detection limit | BDL% |
|---|---|---|---|---|---|---|
| **OC/EC** | | | $\mu g/m^3$ | | | % |
| OC | 20.8 | 17.0 | 89.9 | 1.1 | 0.4 | - |
| EC | 5.6 | 4.4 | 23.1 | 0.2 | 0.1 | 4.3 |
| **SIA** | | | $\mu g/m^3$ | | | % |
| $SO_4^{2-}$ | 23.5 | 20.8 | 95.8 | 0.04 | 0.04 | 0.21 |
| $NO_3^-$ | 22.0 | 23.3 | 104.7 | 0.03 | 0.03 | 0.1 |
| $NH_4^+$ | 14.0 | 14.7 | 66.6 | 0.04 | 0.05 | 1.4 |
| $Na^+$ | 0.39 | 0.32 | 1.89 | 0.02 | 0.04 | 8.5 |
| $Cl^-$ | 4.89 | 4.19 | 27.6 | 0.05 | 0.05 | 0.1 |
| **Metal** | | $\mu g/m^3$ | | | $ng/m^3$ | % |
| K | 1.49 | 1.17 | 5.28 | 0.10 | 2.366 | - |
| Fe | 0.769 | 0.541 | 2.22 | 0.015 | 0.759 | - |
| Ca | 0.384 | 0.277 | 2.08 | 0.001 | 0.902 | - |
| Zn | 0.286 | 0.261 | 1.85 | 0.005 | 0.231 | - |
| Pb | 0.107 | 0.091 | 0.469 | 0.004 | 0.218 | - |
| Mn | 0.058 | 0.046 | 0.210 | 0.001 | 0.283 | - |
| Ba | 0.035 | 0.023 | 0.160 | 0.002 | 0.945 | - |
| Cu | 0.027 | 0.024 | 0.171 | 0.002 | 0.267 | - |
| As | 0.022 | 0.021 | 0.084 | 0.000 | 0.114 | 17 |
| Cr | 0.010 | 0.010 | 0.110 | 0.000 | 0.288 | 11 |
| Se | 0.008 | 0.008 | 0.046 | 0.000 | 0.141 | 5.2 |
| Ni | 0.002 | 0.002 | 0.044 | 0.000 | 0.226 | 3.4 |

* BDL% refers to the percentage of data below the detection limit

\* The unit of the detection limit of each metal is ng/m$^3$

\* '-' means that all data are above the detection limit

**Reviewer #2 Comment NO.11**: *Page 8, line 181: Add a comma after "5 km".*

**Response to reviewer comment NO.11:** A comma has been added after "5 km" (see Page 8, line 199).

**Reviewer #2 Comment NO.12**: *Page 8, line 181: Add a space after "x", and before "2.5"..*

**Response to reviewer comment NO.12:** A space has been added after "x" and before "2.5" (see Page 9, line 202).

**Reviewer #2 Comment NO.13**: *Page 10, line 229-232: you have calculated the concentration of mineral species (Al, Si, Fe) based on Ca concentration, and the composition of urban soil. Dose this typical urban soil was affected by regional and local pollution? During summer or winter? During hazy or non-hazy effect? And what is the estimated uncertainty in this calculation (estimation).*

**Response to reviewer comment NO.13:** The composition of urban soil is investigated by An et al (2016). In this study, 2692 topsoil samples were collected in the urban area of Beijing during 2011. The composition of soil was an annual average result and most of the urban areas of Beijing including our sampling site (PKU) was covered. Based on large amount of samples, the soil composition in this study could be considered as typical urban soil composition of Beijing. Thus we choose it to calculate mineral species in our study. An et al., (2016) identified that local soil in Beijing were influenced by both local and regional sources including industrial, traffic, agricultural and biomass burning source.

Due to the spatial and temporal variability in soil dust sources, it is very difficult to characterize an appropriate aerosol soil composition for a specific site and the uncertainty might be large. The concentrations of Al, Si, Fe and Mg were calculated by the concentration of Ca and the composition of urban soils of Beijing: Al= 1.7Ca, Si= 7.3Ca, Fe=0.7Ca, Mg= 0.3Ca (An et al., 2016). Therefore, the

uncertainty in this calculation could be roughly estimated as follows:

$$\left(\frac{U_{Al}}{C_{Al}}\right)^2 = \left(\frac{U_{Ca}}{C_{Ca}}\right)^2 + \left(\frac{U_{Al/Ca}}{R_{Al/Ca}}\right)^2 \tag{1}$$

$$\left(\frac{U_{Si}}{C_{Si}}\right)^2 = \left(\frac{U_{Ca}}{C_{Ca}}\right)^2 + \left(\frac{U_{Si/Ca}}{R_{Si/Ca}}\right)^2 \tag{2}$$

$$\left(\frac{U_{Fe}}{C_{Fe}}\right)^2 = \left(\frac{U_{Ca}}{C_{Ca}}\right)^2 + \left(\frac{U_{Fe/Ca}}{R_{Fe/Ca}}\right)^2 \tag{3}$$

$$\left(\frac{U_{Mg}}{C_{Mg}}\right)^2 = \left(\frac{U_{Ca}}{C_{Ca}}\right)^2 + \left(\frac{U_{Mg/Ca}}{R_{Mg/Ca}}\right)^2 \tag{4}$$

$$(U_{mineral})^2 = (U_{Ca})^2 + (U_{Al})^2 + (U_{Si})^2 + (U_{Fe})^2 + (U_{Mg})^2 \tag{5}$$

where U refers to uncertainty, C refers to concentration and R refers to ratio. $U_{Ca}$ is the MDL of Ca measured by Xact (0.9 ng/m$^3$) and $C_{Ca}$ is the average concentration of Ca. $U_{Al}$ / $U_{Ca}$ and $R_{Al}$ / $R_{Ca}$ (same with Si, Fe and Mg) are all calculated from An et al., 2016. Based on the above equations, $U_{mineral}$ is 2.46 µg/m$^3$ and the average mineral concentration during the sampling period is 3.28±2.46 µg/m$^3$.

**Response to reviewer comment NO.14:** The sentence has been moved to Page 6, line 150 as follows:

*Chemical closure has been done between the measured and reconstructed PM$_{2.5}$. Organic matter (OM) was calculated as OM= 1.6 ×OC (Turpin and Lim, 2001). Mineral species was calculated as Mineral= 1.89 Al +2.14 Si + 1.4 Ca + 1.43 Fe + 1.66 Mg (Zhang et al., 2003). The concentrations of Al, Si, Fe and Mg were calculated by the concentration of Ca and the composition of urban soils of Beijing: Al= 1.7Ca, Si= 7.3Ca, Fe=0.7Ca, Mg= 0.3Ca (An et al., 2016). Since the concentration of Al and Si were not directly measured by Xact, the calculated mineral component might be underestimated. "Others" were calculated by subtracting OM, EC, Mineral and secondary inorganic aerosol (SIA, including SO$_4^{2-}$,*

$NO_3^-$, $NH_4^+$) concentration from total $PM_{2.5}$ concentration. The correlation of measured and reconstructed $PM_{2.5}$ mass could be seen in Fig. S6 with $R^2=0.892$.

**Reviewer #2 Comment NO.15**: *Page 10, line 234: You have stated that Al and Si might be underestimated. Why? And by how much? Please provide supporting details.*

**Response to reviewer comment NO.15:** As Al and Si are not be directly measured by Xact, the mineral species (including Al, Si, Ca, Fe, Mg) might be underestimated with only Ca and Fe measured. However, we have already calculated the concentrations of Al, Si, Fe and Mg based on the concentration of Ca and the composition of urban soils of Beijing to compensate for the underestimation. Therefore, the sentence "*Since the concentration of Al and Si were not directly measured by Xact, the calculated mineral component might be underestimated*" is no longer correct here and has been deleted in the revised manuscript.

**Reviewer #2 Comment NO.16**: *Page 10, line 241: Here you compare the average OC/EC ratio with Yan et al., 2015. Can you be more specific about the time resolution, duration, months, and/or any special pollution events.*

**Response to reviewer comment NO.16:** Based on the reviewer's suggestion, we add some detailed information about the study of Yan et al., 2015. The time resolution of this study is 1 day (23.5 hours). The sampling period in winter is from January 11[th] to 18[th], 2013. There was a haze episode from January 11[th] to 14[th], with the highest $PM_{2.5}$ concentration of 500 μg/m$^3$. The $PM_{2.5}$ concentration decreased during January 15[th] to 18[th], ranging from 50 ~ 200 μg/m$^3$. Generally, the air pollution during the whole sampling period in winter was severe, with the average $PM_{2.5}$ concentration of 209 ± 145 μg/m$^3$. Therefore, the OC/EC ratio was much higher than that in our study.

**Response to reviewer comment NO.26:** Yes, we agree that this sentence is not necessary and has deleted it from the revised manuscript.

**Reviewer #2 Comment NO.27**: *Page 15, line 271: We can control precursors of secondary sources, but not the secondary sources. Please modify accordingly.*

**Response to reviewer comment NO.27:** We have modified the sentence as follows (Page 16, line 388):

*In the meantime, more control of biomass burning and precursors of secondary source in surrounding areas are also needed to mitigate air pollution in Beijing.*

**Reviewer #2 Comment NO.28**: *Page 20, line 523: Fix (PM2. 5). Extra space.*

**Response to reviewer comment NO.28:** We have changed PM2.5 to $PM_{2.5}$ in the revised manuscript.

**Reviewer #2 Comment NO.29**: *Page 20, line 532: Capitalize the first word of the title only.*

**Response to reviewer comment NO.29:** We have revised the citation as follows:

**Reference**

[1]   Yang, Y., Liu, X., Qu, Y., An, J., Jiang, R., Zhang, Y., Sun, Y., Wu, Z., Zhang, F., and Xu, W.: Characteristics and formation mechanism of continuous hazes in China: a case study in autumn of 2014 in the North China Plain, Atmos. Chem. Phys, 15, 8165-8178, 2015.

**Reviewer #2 Comment NO.30**: *Page 22, line 569: Sulfate and nitrate (check technical comment #21).*

**Response to reviewer comment NO.30:** We have replaced the term "sulfate" with "$SO_4^{2-}$" and the term "nitrate" with "$NO_3^-$".

**Reviewer #2 Comment NO.31**: *Page 24, Figure two: I suggest naming them a and b. Also, please explain the what the white bars represent?.*

**Response to reviewer comment NO.31:** The white bars represent the frequency of $PM_{2.5}$ concentration. Figure 2 has been revised as follows:

[Figure]

**Figure 2.** Variation of (a) chemical composition and (b) elemental species with $PM_{2.5}$ concentration (the white bars represent the frequency of $PM_{2.5}$ concentration).

**Reviewer #2 Comment NO.32**: *Page 26, Figure 4: The right side of the Y-axis shows more than 100%. These are % of what?*

**Response to reviewer comment NO.32:** The right side of the Y-axis shows the concentration of different sources with the unit of $\mu g/m^3$, while the left side of the Y-axis shows the measured concentration of $PM_{2.5}$ with the unit of $\mu g/m^3$. The unit of Y-axis has been revised in Figure 4, Figure 5, Figure 6 and Figure 7.

**Reviewer #2 Comment NO.33**: *Page 27, Figure 5: Move the boxes of PMF source identifiers to the left side of the figure and locate them under source apportionment results only. Also, it would be better if you rename these figures as a, and b.*

**Response to reviewer comment NO.33:** Figure 5 has been revised according to the reviewer's suggestion.

**Reviewer #2 Comment NO.34**: *Page 28, Figure 6: same comment as for (technical comment #32).*

**Response to reviewer comment NO.34:** Figure 6 has been revised as mentioned in Reviewer #2 Comment NO.32.

**Reviewer #2 Comment NO.35**: *Page 29, Figure 7: Check technical comment # 33.*

**Response to reviewer comment NO.35:** Figure 7 has been revised according to the reviewer's suggestion.

**Reviewer #2 Comment NO.36**: *Page 30, Figure 30: Add r and p value for the correlations. And discuss in the text.*

**Response to reviewer comment NO.36:** The r and p value for the correlations have been added in **Figure**

**8** as below. The p for correlations between secondary source and local contribution is 0.022 and the p for correlations between coal combustion and local contribution is 0.036. The p for correlations between biomass burning and local contribution exhibited a possible trend toward significance (p=0.052). The r and p values have also been added in the manuscript as follows (see Page 15, Line 376):

*The results showed that for PM$_{2.5}$ in Beijing, secondary source contribution decreased when local emission was more significant (p<0.05, r=0.4) while coal combustion, as a primary combustion source, showed an increasing trend along with local contribution estimated by NAQPMS (p<0.05, r=0.3).*

[Figure]

**Figure 8.** Correlations of local contribution by NAQPMS with the relative contribution by PMF of (a) secondary source, (b) coal combustion source and (c) biomass burning source.

**Reviewer #3 Comment NO.1**: *Line 52- please add two or three more references for the PM2.5 source apportionment studies. For example you might add the following papers: Kotchenruther, R. a., 2016. Source apportionment of PM2.5 at multiple Northwest U.S.*
*sites: Assessing regional winter wood smoke impacts from residential wood combustion.*
*Atmos. Environ. 142, 210–219.*
*Taghvaee, S., Sowlat, M.H., Mousavi, A., Hassanvand, M.S., Masud, Y., Naddafi, K.,*

*Sioutas, C., 2018. Source apportionment of ambient PM 2.5 in two locations in central*
*Tehran using the Positive Matrix Factorization ( PMF ) model. Sci. Total Environ. 629,*
*Zong, Z., Wang, X., Tian, C., Chen, Y., Qu, L., Ji, L., Zhi, G., Li, J., Zhang, G., 2016.*
*Source apportionment of PM2.5 at a regional background site in North China using*
*PMF linked with radiocarbon analysis: Insight into the contribution of biomass burning.Atmos. Chem.*
*Phys. 16, 11249–11265.*

**Response to reviewer comment NO.1:** Based on the reviewer's comment, the citations have been added in the sentence as follows (Page 1, line 52-53):

*Previous studies have found that $PM_{2.5}$ can be emitted from various sources, including residential coal combustion, biomass burning, traffic-related sources, industrial sources and dust (Gao et al., 2016; Kotchenruther et al., 2016; Taghvaee et al., 2018; Watson et al., 2001; Zong et al., 2016).*

**Reviewer #3 Comment NO.4**: *Line 142- You definitely need to present the average concentration of PM2.5 chemical components in a table for different episodes of your study. This table should also include the min, max, signal/ noise (S/N) ratio for your data as the important parameters in PMF analysis.*

**Response to reviewer comment NO.4:** Detailed parameters of $PM_{2.5}$ chemical components for different episodes have been summarized in Table 4 and Table 5 (**Table S1 and Table S4** in the supplementary materials) as follows:

**Table 4 Statistical summary of identified species of PM$_{2.5}$ in the entire sampling period**

| | Mean | Std. | Max | Min | Detection limit | BDL% |
|---|---|---|---|---|---|---|
| **OC/EC** | | | µg/m$^3$ | | | % |
| OC | 20.8 | 17.0 | 89.9 | 1.1 | 0.4 | - |
| EC | 5.6 | 4.4 | 23.1 | 0.2 | 0.1 | 4.3 |
| **SIA** | | | µg/m$^3$ | | | % |
| SO$_4^{2-}$ | 23.5 | 20.8 | 95.8 | 0.04 | 0.04 | 0.21 |
| NO$_3^-$ | 22.0 | 23.3 | 104.7 | 0.03 | 0.03 | 0.1 |
| NH$_4^+$ | 14.0 | 14.7 | 66.6 | 0.04 | 0.05 | 1.4 |
| Na$^+$ | 0.39 | 0.32 | 1.89 | 0.02 | 0.04 | 8.5 |
| Cl$^-$ | 4.89 | 4.19 | 27.6 | 0.05 | 0.05 | 0.1 |
| **Metal** | | µg/m$^3$ | | | ng/m$^3$ | % |
| K | 1.49 | 1.17 | 5.28 | 0.10 | 2.366 | - |
| Fe | 0.769 | 0.541 | 2.22 | 0.015 | 0.759 | - |
| Ca | 0.384 | 0.277 | 2.08 | 0.001 | 0.902 | - |
| Zn | 0.286 | 0.261 | 1.85 | 0.005 | 0.231 | - |
| Pb | 0.107 | 0.091 | 0.469 | 0.004 | 0.218 | - |
| Mn | 0.058 | 0.046 | 0.210 | 0.001 | 0.283 | - |
| Ba | 0.035 | 0.023 | 0.160 | 0.002 | 0.945 | - |
| Cu | 0.027 | 0.024 | 0.171 | 0.002 | 0.267 | - |
| As | 0.022 | 0.021 | 0.084 | 0.000 | 0.114 | 17 |
| Cr | 0.010 | 0.010 | 0.110 | 0.000 | 0.288 | 11 |
| Se | 0.008 | 0.008 | 0.046 | 0.000 | 0.141 | 5.2 |
| Ni | 0.002 | 0.002 | 0.044 | 0.000 | 0.226 | 3.4 |

* BDL% refers to the percentage of data below the detection limit

* The unit of the detection limit of each metal is ng/m$^3$

* '-' means that all data are above the detection limit

**Table 5 Average concentration of PM$_{2.5}$ and identified species in different haze and non-haze periods**

Unit: µg m$^{-3}$

| Average Conc. | EP1 n=102 | EP2 n=95 | EP3 n=117 | EP4 n=131 | NH1 n=78 |
|---|---|---|---|---|---|
| $PM_{2.5}$ | 97.7±70.7 | 143.8±119.1 | 115.3±108.6 | 241.8±115.5 | 18.8±20.7 |
| OC | 19.1±10.7 | 24.7±18.7 | 23.1±21.1 | 40.3±14.5 | 3.33±2.85 |
| EC | 4.1±2.7 | 7.1±5.1 | 5.6±4.3 | 11.0±3.7 | 0.9±0.6 |
| $SO_4^{2-}$ | 18.6±10.9 | 25.1±20.4 | 23.4±19.7 | 53.3±19.2 | 4.43±3.89 |
| $NO_3^-$ | 19.9±14.6 | 18.9±16.0 | 23.3±23.3 | 56.8±24.8 | 2.81±3.69 |
| $NH_4^+$ | 13.3±8.36 | 12.6±11.6 | 13.3±13.2 | 36.4±16.7 | 2.42±3.80 |
| $Na^+$ | 0.37±0.23 | 0.35±0.28 | 0.49±0.41 | 0.69±0.23 | 0.10±0.14 |
| $Cl^-$ | 5.07±3.05 | 5.17±3.71 | 5.66±5.87 | 8.15±3.41 | 0.97±1.10 |
| K | 2.55±1.33 | 1.21±0.90 | 1.23±1.08 | 2.53±0.78 | 0.410±0.410 |
| Fe | 1.26±0.58 | 0.724±0.568 | 0.641±0.480 | 1.06±0.25 | 0.144±0.133 |
| Ca | 0.530±0.250 | 0.479±0.447 | 0.356±0.249 | 0.360±0.148 | 0.054±0.033 |
| Zn | 0.442±0.242 | 0.242±0.197 | 0.252±0.289 | 0.409±0.170 | 0.060±0.076 |
| Pb | 0.182±0.105 | 0.087±0.071 | 0.090±0.084 | 0.185±0.063 | 0.024±0.028 |
| Mn | 0.093±0.050 | 0.049±0.038 | 0.049±0.045 | 0.082±0.021 | 0.011±0.012 |
| Ba | 0.047±0.016 | 0.038±0.032 | 0.032±0.022 | 0.048±0.012 | 0.006±0.005 |
| Cu | 0.028±0.015 | 0.023±0.021 | 0.028±0.026 | 0.042±0.017 | 0.007±0.009 |
| As | 0.031±0.019 | 0.019±0.019 | 0.021±0.023 | 0.040±0.013 | 0.003±0.006 |
| Cr | 0.019±0.022 | 0.007±0.009 | 0.008±0.011 | 0.014±0.010 | 0.001±0.001 |
| Se | 0.012±0.007 | 0.006±0.005 | 0.007±0.007 | 0.018±0.007 | 0.001±0.001 |
| Ni | 0.003±0.002 | 0.002±0.001 | 0.002±0.002 | 0.004±0.004 | 0.0006±0.0004 |

**Reviewer #3 Comment NO.5**: *Line 149- please add the (Norris et al., 2014; Paatero and Tapper, 1994; Paatero et al., 2014;Paatero, 1997) as the main references for PMF model:*

*Paatero, P., 1997. Least Squares Formulation of Robust Non-negative Factor Analysis.pp. 23–35.*

*Paatero, P., Tapper, U., 1994. Positive matrix factorization: a non-negative factor model with optimal utilization of error estimates of data values. Environmetrics 5, 111–126.*

*Paatero, P., Eberly, S., Brown, S.G., Norris, G.a., 2014. Methods for estimating*

*uncertainty in factor analytic solutions. Atmos.Meas. Tech. 7:781–797. https://doi.org/10.5194/amt-7-781-2014.*

*Norris, G., Duvall, R., Brown, S., Bai, S., 2014. EPA Positive Matrix Factorization (PMF) 5.0 Fundamentals and User Guide.*

**Response to reviewer comment NO.5:** The above references have been added to the introduction of PMF model as follows (Page 7, line 163):

*Factor contributions and profiles were derived by minimizing the objective function Q in the PMF model, which was determined as follows (Norris et al., 2014; Paatero and Tapper, 1994; Paatero et al., 2014;Paatero, 1997):*

**Reviewer #3 Comment NO.6**: *Line 163- Please provide the Q robust values for different PMF solutions in an SI figure. This would be really helpful in showing why you picked the 6 factor solution as the optimal PMF resolved solution.*

**Response to reviewer comment NO.6:** Thanks for the reviewer's suggestion. To determine the optimal number of source factors, a string of effective test, in which factors number was from four to nine, was carried out. The resulting Q parameters were shown in Figure 12 (**Figure S4** in the supplementary materials). Obviously, there was a lowest $Q_{Robust}$ value (13087) at six factors in moving from four to nine factors. Although $Q_{expected}$ has been decreasing in the process, $Q/Q_{expected}$ shared similar variation with $Q_{Robust}$ showing the lowest value at six factors (1.3).

[Figure]

Figure 12. The variation of Q parameters from four factors to ten factors

**Reviewer #3 Comment NO.7**: *Line 166- In addition to briefly touching the results of your uncertainty analysis, you need to mention the uncertainty analysis results in detail (more discussions can be found in PMF source apportionment papers)*

**Response to reviewer comment NO.7:** Uncertainty of PMF model is usually estimated by bootstrapping (BS), displacement (DISP), and bootstrapping with displacement (BS-DISP). Here, characteristics of factors nearby six, where $Q_{Robust}$ was relative lower, were explored. With five factors, three factors were mapped 100% of BS, while industry source and traffic source were mapped 92% and 94%, respectively of runs. There were no swaps with DISP, and 100% of the BS-DISP runs were successfully. At six factors, results were more stable with all factors mapped in BS in 100% (Table 6 below, also Table S2 in the supplementary materials), no swaps occurred with DISP and all BS-DISP runs were successfully. However, the solution became less stable in moving from six to seven factors. The new sea salt factor was only mapped in BS in 87% and coal combustion factor was mapped in BS in 89%, traffic source factor was mapped in 93%, other factors were mapped in 100% of runs. No swaps were found in DISP. Therefore, based on the above analysis, six factors were found to be the optimal solution in this study.

**Table 6 Percentage of BS factors assigned to each base case factor with a correlation threshold of 0.6.**

| Boot Factor | Secondary | Industrial | Dust | Traffic | Coal | Biomass |
|---|---|---|---|---|---|---|

| | | | | | |
|---|---|---|---|---|---|
| 1 | 100 | 0 | 0 | 0 | 0 | 0 |
| 2 | 0 | 100 | 0 | 0 | 0 | 0 |
| 3 | 0 | 0 | 100 | 0 | 0 | 0 |
| 4 | 0 | 0 | 0 | 100 | 0 | 0 |
| 5 | 0 | 0 | 0 | 0 | 100 | 0 |
| 6 | 0 | 0 | 0 | 0 | 0 | 100 |

**Reviewer #3 Comment NO.8**: *—Why the simulation period for footprint model, and NAQPMS model are not the same? For example, the footprint simulation was performed from 1-31 December while the NAQPMS model analysis was performed from 10th of November to 15th of December.*

**Response to reviewer comment NO.8:** The simulation period for footprint model and NAQPMS model are not the same because each analysis was performed by different research group. Based on the input data availability, the footprint simulation was performed from 1-31 December while the NAQPMS model analysis was carried out from 10th of November to 15th of December. Therefore, in this study, we use the data from December 1$^{st}$ to 15$^{th}$ for the analysis when both NAQPMS and the footprint results are used.

**Reviewer #3 Comment NO.9**: *Line 247- Please add a couple of references for the following sentence: In general, the large contribution of SIA, OM as well as the high OC/EC ratio indicated the importance of secondary formation in winter in Beijing, while the high concentration of species like SO42-and K suggested a significant contribution of combustion sources to PM2.5.*

**Response to reviewer comment NO.9:** Several references have been added as follows (Page 11, line 260):

*In general, the large contribution of SIA, OM as well as the high OC/EC ratio indicated the importance of secondary formation in winter in Beijing (Sun et al., 2016b), while the high concentration of species like $SO_4^{2-}$ and K suggested a significant contribution of combustion sources including coal combustion and biomass burning to $PM_{2.5}$ (Achad et al., 2018; Chen et al., 2017; Li et al., 2017b).*

**Response to reviewer comment NO.11:** The detailed explanation of high fraction of K in industrial source and Ni in biomass burning source could be found in Response to reviewer comment NO.5 (Page 9-13 in this response file). Also, in Response to reviewer comment NO.5 and Response to reviewer comment NO.6 (Page 13-16 in this response file), we identified factors and justify our results by comparing the source profiles from PMF results in this study with those of specific emission sources reported in previous studies, and by good correlations between the tracers of identified sources and sources mass concentrations.

**Reviewer #3 Comment NO.12**: *Line 335- How do you compensate the lack of data for regional and local contribution from the NAQPMS model for the EP4?*

**Response to reviewer comment NO.12:** For better understanding of the evolution of EP4, the potential source contribution function (PSCF) model could be conducted to justify the result of the footprint model and compensate the lack of NAQPMS model results. The PSCF model was established by Malm et al. (1986). Total potential source contribution function (TPSCF) model was then developed based on this method by integrating air trajectories from different endpoint heights (Cheng et al., 1993). With combination of pollutant concentration ($PM_{2.5}$) and air mass transport information, TPSCF model was used for analyzing the dominant transport pathways to a certain receptor site (Liu et al., 2017). More detailed information about TPSCF model could be found in Liu et al. (2017). The TPSCF result during EP4 is shown in Figure 13. With higher TPSCF value (in orange, pink and red), the potential contribution to $PM_{2.5}$ at the sampling site (Beijing) increased. From Figure 13, it could be seen that the high TPSCF value concentrated in the southwestern area to Beijing (mostly in Hebei province), indicating that regional

transcript contributed significantly to PM$_{2.5}$ in Beijing during EP4.

[Figure]

Figure 13. TPSCF results during EP4

[revised manuscript text omitted]